# The Lean-Branch-and-Bound Structure Effectiveness in Enhancing the Logistic Stowage Methodology for the Regular Shapes

**Ahmed M. Abed [1,2,*] and Laila F. Seddek [3,4,*]**

[1]  Department of Industrial Engineering, College of Engineering, Prince Sattam Bin Abdulaziz University, Al-Kharj 16273, Saudi Arabia

[2]  Industrial Engineering Department, Zagazig University, Zagazig 44519, Egypt

[3]  Department of Mathematics, College of Science and Humanities in Al-Kharj, Prince Sattam Bin Abdul-Aziz University, Al-Kharj P.O. Box 11942, Saudi Arabia

[4]  Department of Engineering Mathematics and Physics, Faculty of Engineering, Zagazig University, Zagazig 44519, Egypt

[*]  Correspondence: ahmed-abed@zu.edu.eg or a.abed@psau.edu.sa (A.M.A.); l.morad@psau.edu.sa (L.F.S.); Tel.: +96-6509506811 (A.M.A.); +96-6501291418 (L.F.S.)

**Abstract:** An excellent e-commerce logistic cycle is based on reducing the delivery time to satisfy customers, accelerating the distribution chain activities at each delivery station, increasing the transported stowage objects for mobilization parallelograms containers to ingest most orders, and reducing the unused area. Because the stowage steps are considered an NP-complexity, the authors introduce the Oriented Stowage Map (OSM) using one of the heuristic methods (i.e., the camel algorithm) that are programmed by the C-sharp software to be easily managed via the Internet of Things (IoT), which is embedded in the distribution chain. The authors called it Oriented Stowage's Map by Camel algorithm "OSM-CA". This methodology is considered one of the mat-heuristic approaches (i.e., decomposition metaheuristics) because we resorted to using mathematical steps (branch-and-bound). The OSM-CA reduces transport costs by 7% and delivery time by 14%. Additionally, it shows superiority over the solo Ant-colony for stowage less than 50 boxes by 10% and over the solo camel algorithm by 27%, while for more than 50 boxes, the OSM-CA superiority by 30% over the ant colony, and 17% over the camel algorithm. Creating the map in the proposed way takes 70% less time than using mathematical models, especially for a large number of orders, more than 200.

**Keywords:** 2D-packing/stowage; I 4.0; meta-heuristic methods; numerical model; internet of things

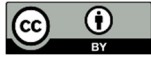

## 1. Introduction

E-commerce aims to fulfill customer requests in the shortest delivery time, without misdelivery. E-commerce is a central store and fleet of moving containers on specific trip paths that serve a sequence of stations (i.e., customers). Every customer receives some products with different dimensions and may return other products as needed. The main challenge is how to stow maximum boxes (i.e., customer requirements) to satisfy the customers by Capital turnover acceleration (i.e., Circular economy). The stowage is a continuous process repeated at every station during the vehicle trip on their transition path by unloading and loading different rectangular boxes with different dimensions. The stowage aims to increase the capacity of the container by increasing the ingested transported boxes at the source station and the ability to receive returned boxes from customers during the trip. The problem may be identified as customizing some boxes in specific best-fit spaces to achieve maximum transported items.

The stowage process achieves mutual interest when transported items increase. The first reduces the transportation cost, and the second is meeting more customer requests, which partially reflects positively on reducing the delivery time. These objectives are the main core of the success of the e-commerce logistic system. Therefore, the stowage issue must be improved. The authors have suggested showing the container floor layout solution in a visual map to simplify the stowage activities during the trip. The first question is, what suitable methodologies help in suggesting the stowage map quickly because the transportation time span between the stations is short?

The authors have resorted to using metaheuristic techniques and chose the camel algorithm to be modified to achieve the main objective (i.e., customize the boxes in best-fit spaces that reduce the unused area of the container layout). The modification is undertaken by supporting the camel algorithm with some mathematical equations to help in precise the best-fit place with the minimum unused area and programming the whole statements of the proposed methodology by c-sharp software, version 2019, structure to create the map visually as will appear in the context and Appendix A.

The second question is: How will the customer-requested products and their returned products (i.e., boxes) deal with the data during the trip? Bearing in mind that e-commerce data are big and rapidly changing. Therefore, the programming of the proposed methodology is important to manipulate all activities through the trip via the cloud and IoT enabling system [1]. The proposed methodology studies the relation of logistic systems, which consists of four elements, e.g., source, products, distribution, and customer satisfaction, as illustrated in (Figure 1), through two sequential stages; the first interested in classifying the objects according to the geometry rules and the second study the transportation cost to measure the efficiency of logistic stowage methodology according to visual lean which help in extract stow map. Therefore, the first half of the proposed methodology is named the oriented stowage map "OSM".

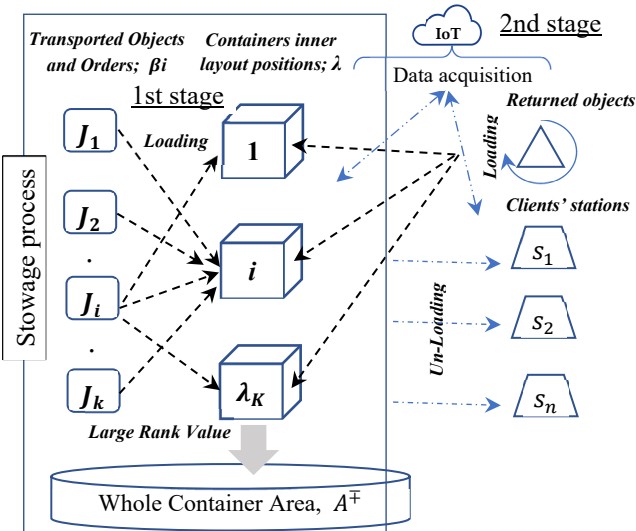

**Figure 1.** The main relationships in the distribution chain need stowage.

The first stage of the proposed methodology has been explained in the next pseudo-code, 'stage-1: Algorithm 1', which describes the main rules and geometrical specifications such as different vertices, centroid, width, length, edges, and fixed container size ..., which are required for the stowage process and will be discussed in Section 2.

---

**Algorithm 1 Stage-1: Stowage Steps Geometrically**

---

**Begin stage-1 do**

**Step 1:** Input a list of objects required to transport to specific customers at a specific station.

> **Step 1.1:** Record the initial specification data of the listed objects (x, y, rotate/not).
> **Step 1.2:** Set all objects where the width (parallel to the x-axis) is greater than their height which is parallel to the (y-axis).
> **Step 1.3:** Rank all objects in descending order according to their width ($\vec{x}$ direction).

**Step 2:** Choose the geometric shape (<u>Rectangular</u>, Irregular, Circular, Polygon);

**Step 3:** Determine the isometric (1D, <u>2D</u>, 3D).

**Step 4:** Identify the stowage orthogonality relation (<u>Yes</u>, or No).

> **Step 4.1:** Determine the degree of freedom *df* (<u>Rotation</u>, Orientation).

**Step 5:** Determine the stowage technique (<u>Guillotine</u>, Non-guillotine).

**Step 6:** Determine the container size $C_{pg}$ (40 ft, 60 ft, customize).

**Step 7:** Manage the GUI interface illustrate in the appendix.

**Step 8: Extract the stowage map according to:**

> **Step 8.1: <u>Station Info.</u>**
>
> > # of served stations in bypass path;
> >
> > # of served stations in backtrack path to collect the (re)turned objects (market policy).
>
> **Step 8.2: <u>Object Info</u>.**
>
> > **Step 8.2.1:** if there a safety boundary around each object (Yes, <u>No</u>)
>
> - **Vertices coordinates**
>
> > **Step 8.2.2:** Choose the first object that have long width in $\vec{x}$ to stow if possible
> > **Elseif**
> > **According to step 4.1 (can Rotate)**
> > **Rotate and stow it if poosible**
> > **Else**
> > **Draw horizontal skyline (guilltine line) Stow the next object over the sky line:**
> >
> > > **Step 8.2.2.1:** Determine their vertices (four vertices for each stowed object).
> > >
> > > **Step 8.2.2.2:** Identify the BLC bottom left placement.
>
> - **Identify the Edges**
>
> > **Step 8.2.3:** Use Eculidian distance between the verticies in same direction $\vec{x}$ or $\vec{y}$ to choose the object that create minimum edge (height difference between two adjacent objects).

**Step 9:** Implement a branch and bound technique aided by the camel algorithm, which searches for a specific target that is the best fit or has the lowest waste area (empty spaces under the horizontal guillotine line).

---

The logistic party focuses on distribution activities, planning, execution, handling/communication, storage, and delivery efficiency. One of the challenges of logistics is the stowage of objects in as little space as possible at the warehouse and/or vehicle container ($\beta$), forcing us to predict accurately consuming products ($\beta'$) areas [2]. The e-commerce system has data characterized by rapid change and bigness. Therefore, the methodology recommends IoT manage this system online after digitizing e-commerce activities and all transported objects and the distribution points (i.e., stations) [3,4]. The IoT is nominated to verify and track the container's objects during loading and unloading until the delivery destination and substitute that every delivered item is offset by a new one requested from the supplier or previous point on the distribution chain [5]. This article addresses many objectives for the proposed methodology, which have been discussed in the stowage problem. The most important of them is to ingest the total stowage transported boxes (i.e., objects) to reduce their cost. They are then reducing the partial delivery time, particularly for objects with time-sensitive freight delivery and in emergency crises when prompt delivery is critical. Additionally, it receives the returned objects during the distribution chain easily [6]. The proposed methodology consists of overlapping steps. The preliminary steps are based on the heuristic technique that helps obtain a semi-optimal solution for ranked objects at a temporary stowage map, while the late steps tackle numerically for enhancing the picked solution from the afore phase and dividing them into the group with the same guillotine boundary [2,6], then trying to replace and switch places between them to ingest all available transported objects. The underlying theory of e-commerce is that the data on all transactions within the warehouse or logistics stock system are digitally tracked through a unified and efficient structure. The IoT provides proactive data during distribution trips to calculate the best utilization for the space caused by unloading objects to be ready for (re)stowing by returned ones, as shown in (Figure 2).

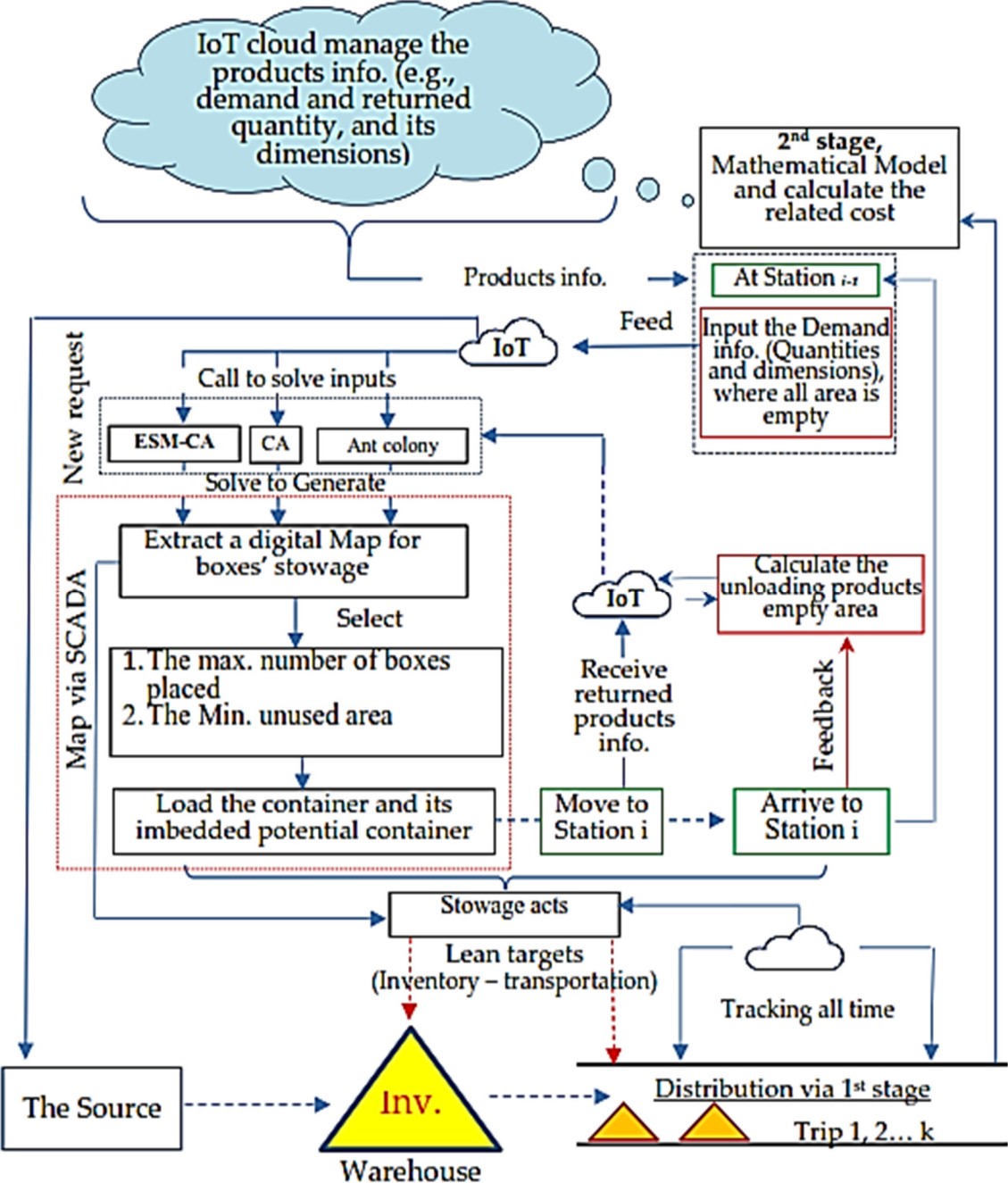

**Figure 2.** The e-commerce framework.

This article's primary objective is to maximize the stowage utilization spaces in the minimum volumetric area to reduce its total transportation cost. The work was triggered by improving the camel algorithm movements over Dana Marsetiya et al. (2022) [7] via supporting the meta-heuristic method by the mathematical procedures that can be programmed by the C-sharp software to enhance the searching process. The authors compare the results with those of Hong J. et al. (2018) and Dana et al. (2022), who studied the effect of the ant colony and camel algorithms on the acceleration of the transportation cycle [7,8].

The article is divided into seven sections, where it tackles the stowage problem geometrically in Section 2 to introduce how the first oriented stowage map aims to exploit all spaces. Section 3 discusses how to verify stowage free of overlapping or any geometric sketch defect by presenting how to measure the space utilization. Section 4 explains the

proposed methodology that will be easily managed via IoT and formulated in Section 5, the results are discussed in section 6. Additionally, it presents an implementation example for transporting bathtubs of different sizes and discusses the results in Section 6. Finally the work expounds the highlight results and their analysis in section 7. The article uses case study data for ceramic tile companies transporting different sizes as appeared in Appendix-A. We finally reviewed the conclusions to show the superiority of the proposed algorithm over the other two optimization algorithms.

## 2. The Stowage Problem Geometrically (Stage-1)

The stowage problem is an open arrangement with a deep and orthogonal axis [5]. This problem relied primarily on the container's floor's geometric dimensions and other characteristics. The fundamental geometric restrictions dominated the stowage problems (i.e., combinatorial optimization problems), where no overlapping of objects in the same layer is allowed [9, 10]. The review reveals that the researchers attempted to develop some methodologies similar to the rank-order stowage problem, where all stowage objects and containers' floors are rectangles. Figure 2 illustrates the structure flowchart (e.g., the constructive heuristic), which is considered the Origin of many modern heuristics. These methods are compared with exact solutions (e.g., LINGO results) [7, 11, 12] by creating the most suitable oriented stowage map for different rectangular-sized products by implementing the camel algorithm. The heuristic steps are supported by mathematical equations [13, 14] to enhance their solutions (i.e., increasing the storage capacity) by following the stowage map created in minimum time. This Guide Map existence must fit, as discussed by Imahori [15], inversely with the speed of the container movement progress and be proportional to the change in demands and/or returns that affect the container's planned spaces and transportation costs. Therefore, the authors propose a methodology called OSM-CA, "Oriented Stowage Map by Camel algorithm" Methodology.

There is an urgent need to create a stowage map for transportation objects during many sequential stations, marine logistics, warehouse capacity, massive stowage of shelves, aiming at loading acceleration, and multidimensional [16,17] resource scheduling. However, compared to the vast benchmarking examples of NP-hard combinatory problems, they are defined by quickly finding appropriate solutions. The "OSM-CA" creates n-branches of solutions to direct the driver to the final minimum stow depth and unused space at any station of their distribution trip. It begins with the ACO that is improved via the OSM-CA heuristic steps, supported by a mathematical model. The OSM-CA has been implemented and managed via an IoT platform to control vehicles, drawers, customers, and warehousing. The camels are responsible for placing boxes in container space under consideration of placing their boxes where their area is proportional to the pungent pheromone scent (i.e., the large box side 'x' | 'y' covers an extensive length of a container). The camels move in descending order on condition, placing their boxes to achieve the best-fit or minimum wasted area case at every guillotine skyline, except that the camels will not find the food and die. The heuristic steps guide the ant movements into multi-stowing branches. The first uses a succession of a horizontal pattern (or vertical) fills only that called "one-step guillotinable layout's floor". While the camels' plan movement that does not have this property is called the "non-guillotine layout's floor" [18–20]. Otherwise, there is permission to orthogonally rotate (i.e., replace the width instead of length and vice versa via rotation 90°) the product that is ordered to stow from the first step and apply the second series [21,22] of guillotine steps (e.g., V or H), as illustrated in (Figure 3). The number of phases necessary to completely cut the pattern provides another way to describe guillotinable designs. The pattern is referred to be "one-stage guillotinable" if all objects may be stowed in the container using only a series of horizontal (or vertical) cuts. The pattern is known as a "two-stage guillotinable pattern" [15] if the orthogonal rotation of the pieces produced by the first stage cuts is required before applying the second set of guillotine cuts (now vertical or horizontal). The same logic also dictates that a third change in cutting direction is necessary for a three-stage guillotinable pattern and

so on to reach no limit of stowage called n phases [23]. In that case, the floor is designated as a "two-step guillotine layout's floor" stowage with no limit on the number of guillotine lines is called n-level guillotinable layouts' floor, where (n-1) is the number of stowage direction changes between $\overrightarrow{x_i}$ and $\overrightarrow{y_i}$. Where $x_i$ denotes the width of the rectangular object needed to be stowed, while $y_i$ denotes their length (or height in 2D).

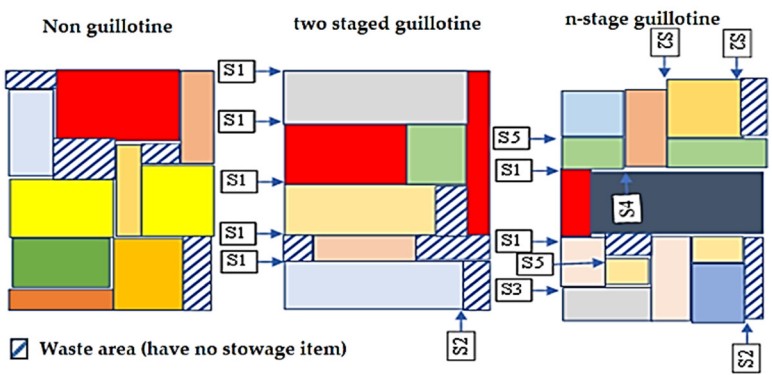

**Figure 3.** The ant plan stowing layout methods.

**Definition 1.** *The dynamic stowage map (e-commerce policy) regularly evaluates the specific objects to be stored in the next period (i.e., the station on the journey path) and produces a potential bespoke container for those demanded and/or returned objects [23].*

## 3. The Problem Scope and Specification

The authors solved 120 benchmarking examples classified in four scales ranging from 10 to 500 transported objects through 2 to 10 available branches for the camel to evaluate the effectiveness of the suggested mat-heuristic "OSM-CA". Suppose the example has more than 200 transported objects. In that case, the authors find that the LINGO solver (native mathematical solution) surpasses the time limit (8–24 running hours) without seeing the best-suggested stowage map in most situations [24]. Table 1 shows the selected size characteristics: The boxes and the capacity of every camel to receive boxes from different branches to place in exact position according to a lagoon surface area (i.e., food acquisition), which is generated by the U (50 and 500) distribution; The same boxes, branches, layout dimensions, and available positions for each problem size [4,25]. The [10,000, 20,000] interval is used to create the range of repositioning cost saving for $\beta_i$ that pass the test. All of the benchmarking examples of the mathematical model and the other three metaheuristic methodologies ACO [13], CA, and OSM-CA are solved and implemented by laptop core I5 twelve generation via processor 2.46.

**Table 1.** Instances the sizes generated randomly.

| Problem Sets | No. of Instances | No. of Branches | No. of Camel Capacity | No. of Available Positions/Boxes |
|:---:|:---:|:---:|:---:|:---:|
| $b_{G1,}$ | 30 | 2 | 5 | 10 |
| $b_{G2,}$ | 30 | 4 | 8 | 50 |
| $b_{G3,}$ | 30 | 6 | 10 | 100 |
| $b_{G4,}$ | 30 | 10 | 20 | 500 |

## 4. The Proposed Methodology OSM-CA

The dynamic stowage problem is introduced in this section. This debate concerns a group of objects that must be transported along a specific path by a particular vehicle. The camel metaheuristics algorithm resorts to looking for lagoons' positions in the container over four well-known strategies such as "positioning", "fitness", "level-plane", and

"profile". While the numerical (i.e., exact) algorithm guarantees optimal solution, it may consume much time, generate exponential iterations of multi-branches [21, 26], and be very hard if programmed. Whereas the heuristic steps, such as the greedy method [27], local search algorithms as constructive steps [28], or metaheuristic, as ACO and CA [4, 29] find a most favorable, i.e., $m_f$, solution quickly, its accuracy is not substantively. The prominent exact algorithms are the cutting plane [30], branch-and-bound, branch-and-cut [31, 32] and dynamic programming methods [33]. The proposed methodology, "OSM-CA", is based on a modified branch and bound heuristic steps to support the CA to obtain a local $m_f$ solution and then enhance it by mathematical procedures to guarantee efficiency. Additionally, the authors exclude the cutting plane method despite being fast because of unreliability, while the branch/bound way is steadfast but consumes a long time to extract the solution.

Accordingly, the authors enhance the CA and branch-and-bound technique, especially after mathematical procedures. The pre-step of the first phase aims to prepare the problem by classifying objects $J_j$ (i.e., boxes/orders) according to the ratio ($r_i$) among their $x_i$ and $y_i$, which assumes that all boxes with r < 0.2 aside are roomed in the potential container side. The first aims to place boxes, which is achieved using a multilevel linked data structure (i.e., matrix of all boxes bi ($x_i$, $y_i$)). At the same time, the container's width is constant at the beginning of the proposed methodology and is called "$X_{avi}$". Available length is the key to selecting an appropriate box *bi* and permits the camel to place its *bi*. The fitness value, in this case, is not only based on the minimum Euclidean distance between adjacent boxes on both sides with the centroid of the grouped boxes to reduce the number of edges by measuring an angle that occurs when a correlation triangle appears, as discussed in (Figure 4) above. However, also try to make the skyline close to horizontal as acute as possible.

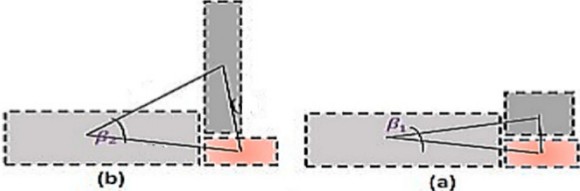

**Figure 4.** Enclosing some of boxes interspersed with unused space as a one block $B_i$.

## 5. Problem Formulation

The objective is to ingest all e-requested orders (i.e., the boxes), which can be described as follows: a set of camels (*n*) responsible for moving the selected boxes via a specific camel $B_{i=1}$, $i = \{1, 2, \ldots, i, \ldots, B_{i=n}\}$, a set of generated branches $S$ for every repositioning under condition (fitness: Is wasted space reduced?), $J = \{1, 2, \ldots, j, \ldots, f_s\}$, and a set of $p$ suitable replacement positions, $K = \{1, 2, \ldots, k, \ldots, p_p\}$. The model assumptions modified for that which is cited in (Hong et al., 2018) [7] when interested with costs. Any placement layouts' floor of boxes considered a set of boxes devoted to each box's corners (four vertices and centroid) and the container and concerning the following assumption in tackling the stowage problem [33]. A set of objects will be transported on an available vehicle container for a given time, passing through Clients' stations. These objects form $B$ disjoint groups, $b_{G1}, b_{G2} \ldots, b_{GB}$. The objects in a group require the same client and transportation means (i.e., drawer). Because the stowing of one group does not affect the stowing of another, the problem may be broken down into sub-problems for each group. As a result, the remainder of this essay will focus on one group's issue. The authors tend to discuss the stowage problem as some objects seek the best-fit location to place via an actuator (camel, ant, etc.) that is proposed to be a camel according to the camel algorithm. The goal of the task is to maximize the overall profit from the box assignment, which is

inversely related to the transportation expenses (e.g., involves the cost of health exhaustion, which implicates time consumption in front of the lagoon). Even though it does not follow the time windows, the OSM is regarded as a soft time window depending on the container receiving the box (best-fit case). The mathematical model for the matheuristic OSM approach was developed based on Ali et al., 2022 [34]'s implementation of the camel algorithm 'CA.' This approach improves the stowage outputs map that has been created. Accordingly, the article is based on (Hong et al., 2018)[6] for testing the cost indicator while based on Dana Marsetiya Utama et al., 2022 [7] Ali et al., 2019 [34] for testing the area indicator, which is hands are proportional directly together and considered the basis of the proposed algorithm. Table 2 indicates the parameters, variables, and decision variables used in the following procedures.

**Table 2.** The Parameters of OSM-CA algorithm constraints.

| **Indices** | |
| --- | --- |
| $i, j$ | Index for lagoon location at a specific branch |
| $S_i$ | Boxes $i$ capacity |
| $SC_j$ | Branches for different $j$ Storage capacity |
| $D_k$ | New Position $k$ new demand |
| **Variables** | |
| $K$ | Total Camels |
| $L$ | Total lagoons location |
| $d_{ij}$ | Distance from lagoon $i$ to lagoon $j$ (i.e., the distance between centers of two adjacent objects) |
| $LCT$ | The cost of poorly choosing a branch of the path when searching about lagoons (cost/time) |
| $Cf$ | Fuel prices; the cost of health exhaustion |
| $LPK$ | Rate of exhaustion consumption per distance |
| $L_k$ | Load time of camel $k$, which is directly proportional to the box area (object area) |
| $S_j$ | Arrival time of the camel to suitable lagoon $j$ |
| $S_i$ | Arrival time of the camel to suitable lagoon $i$ |
| $w_i$ | Waiting time of the camels at the lagoon to place their box |
| $V_{e_{ij}}$ | Camel the fitness effort from lagoon $i$ to lagoon $j$ |
| $ST_{ik}$ | Eating time of lagoon grass $i$ by camel $k$, which is also directly proportional to the box area. |
| $ET_j$ | Permitted time of moving the camel to lagoon $j$ |
| $LT_j$ | Closing time at lagoon $j$ |
| $q_k$ | The capacity of camel $k$, which is directly proportional to the box area |
| $b_i$ | The demand requires from lagoon $i$ according to available place at the container |
| $TTC$ | Total assigning profit of boxes, which is inversely proportional to transportation costs |
| $p_i$: | The adjacent box that will be stowed aside in the one of a degree of freedom (right, left, above, bottom, front, behind) |
| $\beta_i$: | Some of the adjacent boxed boxes are ready to be replaced set, $i = 1,2,\ldots,m$ |
| $J$ | # of Branches set, $j = 1,2,\ldots,S$ |
| $C_{pg}$: | Potential container => $min$ (Space $(b_{i,i+1})$ – [Space $(item_i)$ + Space $(item_{i+1})$]) = The Hype about the pheromone scent |
| $K$ | Another New Position set, $k = 1,2,\ldots,p$ |
| $b_i(x_i.y_i)$: | The box dimension according to container space coordinates |
| $\Omega_b$: | The aggregated boxes interspersed with unused area |
| $r_i = {}^{x_i}/y_i$ | The ratio between the width $w_i = x_i$, while the length Li = $y_i$ of $b_i(w_i.L_i) \leq 0.2$ |
| $W_j(X_j.Y_j)$: | The primary container dimension, where $X_j$ is constant, and $X_j = X_{available\ (0)}$ at the first Stowage step. While $Y_j$ Is extended with Stowage operation and unlimited. |
| $w_k$: | The potential container matrix chain (fleet of camels) is based on the $r_i$ classification. |
| $W_{origin}(0.0)$ | The origin (0, 0) is at the bottom left corner (BLC) of the stowage container. |

| | |
|---|---|
| $C_{bi}$ $(0.5\,x_i.\,0.5y_i)$: | The centroid of the box $b_i$ to calculate the Euclidian distance between $b_i$ and neighbor $b_{i+1}\|b_{i-1}$ |
| $x_{ij}^s\,(x_i.\,y_i)$: | The vertex coordinates of stowed boxes $b_i$ to determine the skyline polygon, |
| $w_i^s \cong x_i^d\|y_i^d$: | Is the width of the selected box preferred to be assigned, with freely rotating the item if it will serve the solution? |
| **Decision variables** | |
| $x_{ijk}$ | A binary variable that controls the movement from lagoon i$^{th}$ to lagoon j$^{th}$ by the Camel k |
| $y_{ki}$ | A binary variable describes the Camel $k$ drinking from lagoon $i$ or discover it mirage. |
| $xd^i$ | Formulates the location for camel $i$ in their vector, where $i$ = 1, 2, …, N, and $d$ = 1, 2, …, D. |

Table 2 shows the parameters used in the proposed methodology 'OSM-CA'. The geometric attributes of boxes include dimensions, physical, weight, volume, color, direction, etc.:

**Assumption 1.** *The container contains multi-reciprocity potential containers called drawers.*

**Assumption 2.** *Any two objects loaded cannot overlay in the container.*

**Assumption 3.** *The boxes' assignment must be orthogonal, i.e., the sides or edges must be parallel to the container side as discussed in (Figure 5).*

**Assumption 4.** *Any box in the container may be easily rotated by 90 degrees [33].*

**Assumption 5.** *As seen in the diagram illustrated in (Figure 6), boxes can be stacked on top of each other with some constraints (durability factor).*

The time it takes to prepare a vehicle for non-stowing and stowing objects is $u^0$ and $u^1$, respectively. Let $D_L$ represent the direct labor cost (USD/h) for the preparation and packing time. When a client order $J_j$ requires $q_j$ an item type, $i_j$. Each type of item's $i_j$ occupied area, $A_s^0(i_j)$, which contains the essential inter-object spacing $\cong 0$ in this study (as shown in Appendix A). Let $O_j = q_j A^0(i_j)$ be the total occupied area for a client's order $J_j$ (objects/client). (We presume that all areas in some unit of measurement may be written as integers). All objects must be stowed if possible. Therefore, every camel seeks the best fit area along time t to customize the selected object $j_j$ elected from the matrix $w_0$, which ranked objects in descending order. If customizing is impossible, the object is stowed temporarily in a potential container $C_{pg}$ and reselect in the final customization stage in time $t_j^0$. (i.e., demand distributed during trip path). The expense of delivering each packing order to its designated drawer is $c_j^0$. Let $F_j = N_j^0(c_j^0 + t_j^0 D) + u^0 D$ be the total of a packing order and prep cost of not stowing an order $J_j$, at one drawer or not completely stowed (the cost of holding no orders or fail the Camel to find the suitable lagoon). Now think about group stowing, $b_G$. Let $X_j = 1$ if order $J_j$ is in the specific drawer/client, and $X_j = 0$ otherwise. While in the case of stowing objects, let $Y = max_{j \in G}\{X_j\}$., i.e., tends to achieve the object $Y = 1$, otherwise $Y = 0$. Let $S = u^1 D$ be the vehicle prep cost to transport particular objects in the same drawer. Let $N(X)$ be the positions area required according to the stowing decision, where the total area of the stowed objects and the empty, serviceable area is $A$ (i.e., whole container area). $N(X) = \left\lceil \frac{\sum_{j \in G} O_j X_j}{A} \right\rceil$, where$[x]$ (the order's number of pieces divided by the drawer's number of parts). While the time required to load returned objects (backing orders) is $t^1$ and the cost of it is, $c^1$. Let $R = c^1 + t^1 D$ be the total cost of loading returned orders. Formerly, for an agreed stowing decision,$X$, the full prep and packing drawer cost for the group is the following sum:

$$C(X) = \sum_{j \in G} F_j(1 - X_j) + \left[\frac{\sum_{j \in G} O_j X_j}{A}\right] R + YS \; \forall j \in b_G \tag{1}$$

The non-negative integer, $s_j$ ; $\sum_{s_j=1} A \le O_j < \sum_{(s_j+1)} A$. Then, the wasted area value, $e_j$, $O_j = \sum e_j + \sum_{s_j=1} A$, and $0 \le e_j < A$. Where the $s_j$ is the number of whole returned orders $J_j$ that requires $e_j$ (i.e., the extra spaces that the returned order, $J_j$, requires transported). Formerly, the authors can write the dynamic fitness function of searching the best stowage orders, as shown in Equation (2). At the same time, the challenge is to stow most boxes carried to reduce the total transportation cost (TTC), which is expressed in Equation (3). It is divided into two sections: the first discusses exhaustion consumption per distance, which is a cost of the searching process, and the second discusses the most expensive branch selection costs.

$$A \propto C(X) = \sum_{j \in G} F_j + \sum_{j \in G}(S_j R - F_j)X_j + \left[\frac{\sum_{j \in G} e_j X_j}{A}\right] R + YS \tag{2}$$

$$A \propto TTC = \sum_{i=0}^{L}\sum_{j=0}^{L}\sum_{k=1}^{K} Cf. LPK. d_{ij}. x_{ijk} + \sum_{j=1}^{L}(max(0, (S_j - LT_j))). LCT + C(X) \tag{3}$$

Subject to:

$$X_j \le Y \; \forall j \in G \tag{4}$$

$$X_j \in \{0,1\} \; \forall j \in G \tag{5}$$

$$Y \in \{0,1\} \tag{6}$$

This problem is formulated as an NP-complexity problem, especially if all of the prep and extra drawer costs are zero. It is worth noting that, despite the extreme integer points, the linear programming relaxation does not yield an optimal solution because the goal function is nonlinear.

$$\sum_{i=0}^{L} g_i \cdot y_{ki} \le q_k, \forall k = 1,2,\dots,K \tag{7}$$

$$\sum_{K=1}^{K} y_{ki} = 1, \forall i = 1,2,\dots,L \tag{8}$$

$$\sum_{K=1}^{K} y_{kO} = K \tag{9}$$

$$\sum_{i=0}^{L} x_{ijk} \le y_{ki}, \forall j = 0,1,2,\dots,L; \; \forall k = 1,\dots,K \tag{10}$$

$$\sum_{i=0}^{L} x_{ijk} \le y_{ki}, \forall i = 0,1,2,\dots,L; \; \forall k = 1,\dots,K \tag{11}$$

$$L_k + \left(s_i + w_i + ST_{ik} + \frac{d_{ij}}{Ve_{ij}}\right). x_{ijk} = s_j, i = 0,1,2,\dots,L; \; j = 0,1,2,\dots,L; \forall k = 1,\dots,K \tag{12}$$

$$w_i = max\,(0, (ET_j - s), i = 0,1,\dots,L. \tag{13}$$

$$x_{ijk} \in [0,1], i = 0,1,2,\dots,L; j = 0,1,2,\dots,L; \; \forall k = 1,\dots,K \tag{14}$$

$$y_{ki} \in [0,1], i = 0,1,2, \dots, L; \ \forall k = 1, \dots, K \tag{15}$$

Equation (7) stipulates that transporting whole boxes must not surpass the camel's carrying capability. According to Equations (8) and (9), each box must have a delivery service. As stipulated by Equations (10) and (11), each order's service can only be fulfilled by a single camel: The time window limitations are defined by Equations (12) and (13). Finally, Equations (14) and (15) specify that the decision variables $x_{ijk}$ and $y_{ki}$ are both binary amounts. Algorithm 1 contains the ***pseudo-code*** for the proposed OSM-CA methodology. The first interests in identifying the procedures of the camel positions' configuration at the BLC (bottom left corner); the second conversion of the camel positions to the trip sequence using the Large Rank Value (LRV) technique to sort the *Rand* and $xd^i$ in descending order; and final, trying to modify their locations as discussed in Eqn. (16-34).

$$x'_{b,k} + x'_{k,b} + y'_{b,k} + y'_{k,b} + z'_{b,k} + z'_{k,b} \geq \Omega_b + \Omega_k - 1, \forall b \neq k \in B \tag{16}$$

$$X_b + x_b \leq X_k + L \cdot \left(1 - x'_{b,k}\right), \dots \forall b \neq k \in B \tag{17}$$

$$Y_b + y_b \leq Y_k + W \cdot \left(1 - y'_{b,k}\right), \dots \forall b \neq k \in B \tag{18}$$

$$Z_b + z_b \leq Z_k + H \cdot \left(1 - z'_{b,k}\right), \dots \forall b \neq k \in B \tag{19}$$

When the variables $x'_{b,k}, x'_{k,b}, y'_{b,k}, y'_{k,b}, z'_{b,k}, or\ z'_{k,b} = 1$, the two boxes surrounded by boxes $b$ and P$_k$ must not overlap with other boxes, as discussed in Section 2 of the article. To avoid the surrounded boxes occupying the same portion of space, it is adequate for no overlying along at least one relative stowage position.

### *5.1. Step-1 (Stowage Boxes) Procedures*

Initially chosen, the stowage layouts of the orthogonal boxes in the containers; an arrangement representation map developed by the authors, addressed enhancing the stowage methodology map and managing it via IoT "OSM-CA". The following part goes through a general description of the recommended technique and an example explaining its sequential steps (Figure 5) in a standard case.

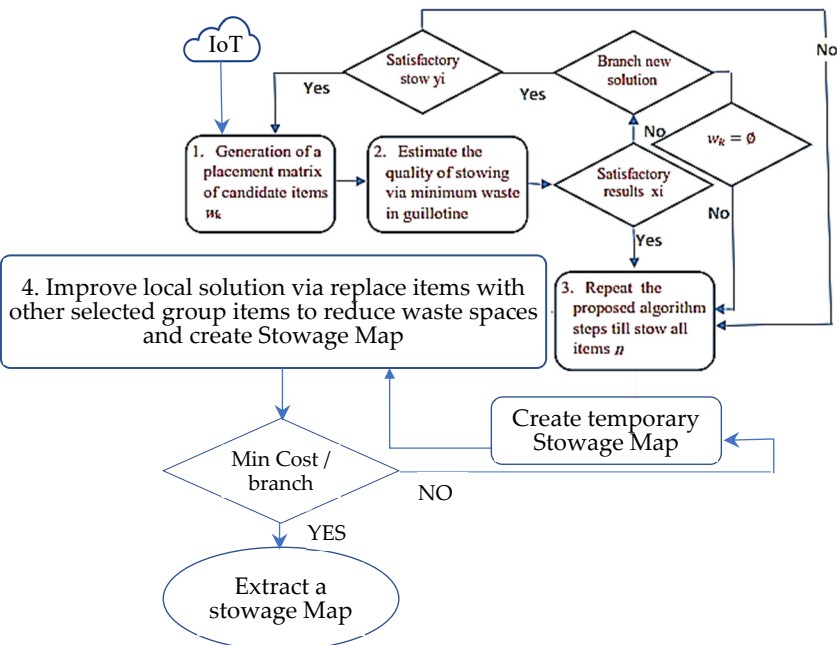

**Figure 5.** The OSM-CA flowchart.

$$\forall \{W_j^1, W_j^2 \dots\} \in W_j^D \, j \in \{1. \dots \dots N\} \tag{20}$$

$$\forall \{\forall \{b_i^1, b_i^2 \dots\} \in b_i^D\} \in W_j^D \, i \in \{1 \dots \dots n\} \tag{21}$$

The proposed procedures aim to find a suitable placement of all boxes that have been candidates into the fewest number of trials under the next circumstances of correct placement acts:

(1)  All vertices of boxes' $\in$ potential container $C_{pg}$, and the main container are parallel,

(2)  All stowed boxes do not overlap with each other, i.e.,

$$\forall j \in \{1 \dots N\}. \forall d \in \{1 \dots D\}. \forall i. k \in \{1 \dots n\}. i \neq k \tag{22}$$

$$\left(x_{ij}^d \geq x_{kj}^d + w_k^d\right) \cup \left(x_{kj}^d \geq x_{ij}^d + w_i^d\right) \tag{23}$$

(3)  All stowed boxes are within the bounds of the containers, i.e.,

$$\forall j \in \{1 \dots N\}. \forall d \in \{1 \dots D\}, \forall i \in \{1 \dots n\} \tag{24}$$

$$(x_{ij}^d \geq 0) \cap (x_{ij}^d + w_i^d \leq W_j^d), \text{ and the } W_j^d \geq X_{available}^d$$

$$Container\ W_i \subset potential\ containers\ P_i \subset boxes\ b_i. \tag{25}$$

Where the $x_{ij}^s$ is the selected coordinate of the assigned vertex, whereas $w_i^s$ is the width of the selected box $b_i$ to be assigned.

$$L_P = \{P_1; P_2; \dots; P_D\}. P_d \in [1; D] \forall d \in \{1 \dots D\} \dots \dots \dots \tag{26}$$

All regular shape classifies according to $(r_{i.x} | r_{i.y}) \rightarrow then\ Stow\ (x_i.y_i) direction.$

(4)  If $(r_{i.x} | r_{i.y}) \leq 0.2$ then $b_i$ reside sticking right or left of container edges around its perimeter rotate the $b_i$ 90° at the bottom of the container to make $\beta = 0$ on the horizontal line. The $X_{avi}$ = Euclidean distance between two inner boxes in the same guillotine layer.

(5)  Check all $X_{avi}$. This is different from the guillotine layer above, considering that the stowage space is the available inner free space in the main container.

A random example of any dimensional stowage problem is represented by a location $w_k$, which holds a sequence of boxes selected for placing into the container, this $w_k$ is a matrix that has some of the box's attributes such as; width $x_i$, length $y_i$, centroid $c_i$, and the ratio between the width and length $(x_i/y_i = ri_{\overrightarrow{x}})$ [35], heights and their weights to classify the boxes before being stuck with container sides. Generating these matrices results in many branches of the solutions, which must be followed to stop at one of two decisions; the first is to create a sub-matrix and continue finding the solution or reach an empty matrix if it has *-ve* or $\emptyset$ values. The sub-matrix considered a branch-and-bound behavior via adopting some of the potential containers (PC, such as drawer; $C_{pg}$) placed in a container at a specific point but ignored the core relationship between the boxes' dimensions and their boxes [36]. Additionally, it missed the skyline path (i.e., an imaginary polygon bounded by the upper vertices of placed boxes). Each PC; $C_{pg}$ with a number, $k$ is described with an objected-vector containing dimensions of this box and adjacent to the BLC, which holding $\{p_k^1; p_k^2; \dots; p_k^D\}, \{x_k^1; x_k^2; \dots; x_k^D \leq X_{available}^W\}$.

In a container, all existing free orthogonal spaces are defined as a set of boxes. The steps of OSM-CA guarantee the correct placement of a box if this box overlays no borders of the potential place $C_{pg}$ in which it is positioned. In this case, a box is put at some point coordinates instead of inspecting the intersection, using a matrix test and stowing everything in its place [37,38]. This test means compromising among the camels to choose the

most appropriate [33]. The rule of correct placement for box $bi\ (xi.yi)$ in a potential place $k$ is expressed with the inefficient:$\left(w_i^d \le p_k^d\right) \forall d \in \{1 \ldots D\}$, as illustrated in (Figure 6).

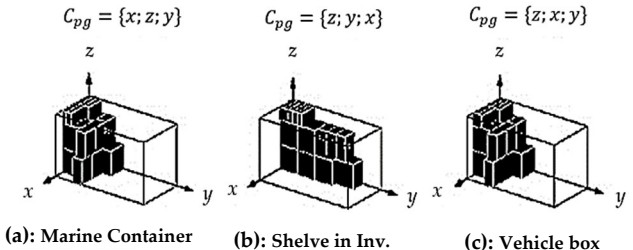

**(a): Marine Container**    **(b): Shelve in Inv.**    **(c): Vehicle box**

**Figure 6.** Load of potential orthogonal container (drawer).

(6)    A set of $\beta$ boxes with the dimensions

$$\{p_k^1; p_k^2; \ldots; p_k^{d-1}; x_i^d - x_k^d; p_k^{d+1}; \ldots. p_k^D\} \tag{27}$$

Additionally, located at an origin of coordinates of the original potential container $k$:

$$\left\{x_k^1; x_k^2; \ldots; x_k^d; \ldots; x_k^D\right\} \tag{28}$$

Which are generated under the following overlap constraints:

$$x_i^d > x_k^d \ and \ x_i^d < x_k^d + p_k^d \ \forall d \in \{1 \ldots D\}; \tag{29}$$

(7)    A set of $\beta$ boxes with the dimensions

$$\{p_k^1; p_k^2; \ldots; p_k^{d-1}; x_k^d + p_k^d - x_i^d - w_i^d; p_k^{d+1}; \ldots. p_k^D\} \tag{30}$$

Additionally, located at $\beta$ points vertices with coordinates

$$\left\{x_k^1; x_k^2; \ldots x_k^{d-1}; \ldots; x_i^d + w_i^d; \ldots; x_k^{d+1}; \ldots; x_k^D\right\} \tag{31}$$

Which are produced under the following restrictions of overlap:

$$\{x_i^s; w_i^s > x_k^s \ and \ x_i^s + w_i^s < x_k^s + p_k^s \ \forall d \in \{1 \ldots D\} \tag{32}$$

$C_{bi}(0.5\ x_i. 0.5y_i)$: The centroid of the box $b_i$ to calculate the Euclidian distance between $b_i$ and neighbor $b_{i+1}|b_{i-1}$

$w_i^s$: is the width of the selected box to be assigned

Illustrative Example (The Positions' Configuration)

The camel's location is determined by adjusting several parameters used in the camel's configuration phase, such as the total camel caravan (N), traveling iterations (iter), maximum and lowest temperatures, and visibility value. The BLC defines the starting position at every trial (d) of the camel's location in this study as determined by the number of problem nodes (lagoons). Equation (33) demonstrates how to pick the camel location's configuration. The camel's starting location is chosen randomly from some camel caravans and nodes (D). The camel's value's upper and lower limits are computed. The camel caravan location's highest maximum is called, $x_{max}$. The camel caravan location's bottom limit is called, $x_{min}$. *Rand* is a uniformly distributed random number in the range [0,1]. The $xd^i$ is calculated as follows:

$$xd^i = (x_{max} - x_{min})Rand + x_{min} \ \ldots \tag{33}$$

$$population^{Accept} = \begin{bmatrix} 1.21, 3.92 & 1.71 & \cdots & 2.18 \\ & \vdots & \ddots & \vdots \\ 8.57, 2.99 & 7.56 & \cdots & 7.94 \end{bmatrix} population^{rej.} = \begin{bmatrix} \begin{bmatrix} 2.18, 3.92 & 1.18 & \cdots & 1.18 \\ & \vdots & \ddots & \vdots \\ 8.57, 8.57 & 7.56 & \cdots & 7.94 \end{bmatrix} \end{bmatrix} \cdots \quad (34)$$

The vector position of 10 nodes and 10 camel caravans is shown in the detected population. At this point, the camel $i$ ensures that each $d$ does not repeat itself. The camel population can be approved if each camel does not have the same quantity in one population (different lagoons and/or boxes). As a result of the application of LRV, the camels will be guided to sort their boxes according to the lagoon area, as illustrated in Table 3. LRV is a straightforward way for converting a pair of (centroid, area) into a Stowage method behavior combinatorial issue. As a result, the serial numbers of camel places must be converted into a traveling sequence; this method is well-known for this purpose. The LRV concept is to sort the d positions of each camel in ascending order. The IoT is a dynamic dataset that manages the generated OSM solutions, which is programmed by C# for all candidates' boxes in the logistic cycle to cope with unloading action and returned objects online on a specific trip [4,39].

1. Create the main matrix $w_k$ of all boxes $b_i$ have permission to stow in the container according to the (Figure 7) structure.

    1.1. Classify w.r.to $xi/y_i = ri_{\underset{x}{\rightarrow}}$ & $xi/y_i = ri_{\underset{y}{\rightarrow}}$ and select the matrix $w_k^{\#1|\#2}$ that

    have # of $b_{i \geq \widehat{x}|\widehat{y}}$, and the average calculated via $\sum_1^n Area(b_i)/Area(C_{pg})$

    1.2. Stowage the perimeter boxes $b_i$ In CCW direction and ignore the container's topside because of its set height assumption as agreed.

    1.3. Calculate the $X_{avi}$ as aforementioned in Step 7 of a proposed methodology, for the illustrative example step shown in Table 3.

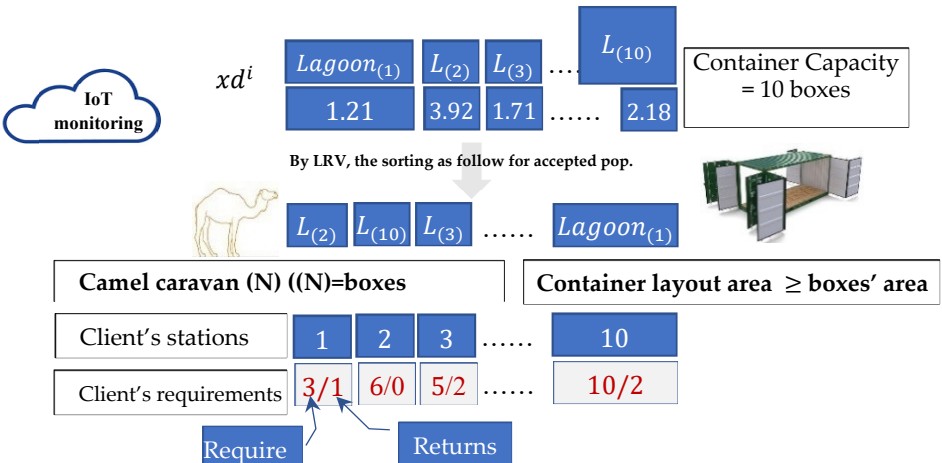

**Figure 7.** The OSM-CA plan representation.

**Table 3.** The main matrix of orders information at station i-1.

|        | b1   | b2   | b3   | b4  | b5   | b6   | b7  | b8  | b9  | b10  | b11  | b12 | b13 |      |
|--------|------|------|------|-----|------|------|-----|-----|-----|------|------|-----|-----|------|
| $x_i$  | 70   | 50   | 70   | 40  | 90   | 40   | 30  | 25  | 25  | 15   | 80   | 12  | 10  |      |
| $y_i$  | 20   | 30   | 5    | 20  | 5    | 35   | 15  | 5   | 10  | 10   | 3    | 10  | 10  |      |
| $A$    | 1400 | 1500 |      | 800 |      | 1400 | 450 | 125 | 250 | 15   |      | 120 | 100 | 6295 |
| $r_i$  | 0.29 | 0.6  | 0.07 | 0.5 | 0.08 | 0.87 | 0.5 | 0.2 | 0.4 | 0.67 | 0.04 | 0.84| 1   |      |

1.4. Stowage $\forall\, b_i$ that have $r_i{<}0.2$ in a separate matrix $w_k^{\#1}$ around the perimeter sides of the container and explained in Table 4.

**Table 4.** Matrix of $w_k \rightarrow r_{i<0.2}$.

|  | $b_3$ | $b_5$ | $b_{11}$ |
|---|---|---|---|
| $x_i$ | 70 | 90 | 80 |
| $y_i$ | 5 | 5 | 3 |
| $C_{bi}$ | (1.5,40) | (2.5,45) | (2.5,35) |
| $r_i < 0.2$ | 0.07 | 0.056 | 0.038 |

2. Create a matrix of the potential container, $w_k^{\#2}$, which have all boxes $b_i$ ready to stowed.

   2.1. If any $b_i$ have many objects $\delta$, collect them besides to make a group as one box have maximum width $= X_{avi}$ and if there is some not collected to this box, create a new box till its width $\sum_{j=b_i}^{\delta} x_i \leq X_{avi}$ (this constraint if all objects $\delta$ belongs to one client).

3. Set all $b_i(x_i.y_i)$, where $x_i \geq y_i$.

4. Arrange all $b_i$ in descending order according to $x_i$.

5. Determine the origin of the main container $W_{origin}(0.0)$, which is preferred at the bottom left corner (BLC) [40].

6. Determine the ratio between the width and length of $b_i(x_i.y_i) \cong r_i = (x_i \,/\, y_i)$, and classify all $b_i$ that its $r_i$ less than in $\boldsymbol{Q}_{1.first\ quartile}$ $matrix$.

7. Determine all vertex coordinates $x_{ij}^s\ (x_i.y_i)$ of the stowed boxes $b_i$ to determine the skyline polygon in CCW order for the free space side. Where; $x_{12}^d(0.0)$ means that vertex of box #1 at container #2 placed one of their vertices at (0, 0) and $x_{12}^{v1}(0.y_i)$ determine the second vertex in the same horizontal edge; by this way, select the other vertex coordinates.

8. Assume that $X_{available\ (0)} = X_j$ at the first stowage step. Otherwise, determine $Min(y_i)$ for these vertices and calculate the $X_{avi\ (1)} = \left| x_{12}^{v2}(0.y_i) - x_{12}^{v4}(x_i.y_i) \right|$.

9. Create the $w_k$, which represents a potential container matrix that contains all $b_i$, ready to stowed/arrangement and consider this matrix a formed of chromosome chain, which is a key to generating a suitable layout's floor.

   $if\ X_{avi} \geq\ \forall x_i\,|\,y_i in\ w_k\ of\ b_i$

      $then\ search\ \exists\ \forall w_k\ to\ compute\ wastage = X_{avi} - x_i \leq 0.$

      $(zero\ case\ is\ best\ fit\ case)$

      $select\ the\ \max wastage\ and\ set\ it\ as\ new\ X_{avi(1-i)}$

      $if\ else\ X_{avi} - y_i \leq 0.\,(zero\ case\ is\ best\ fit\ case)$

   $if\ there\ is\ two\ or\ more\ x_i equals,\ select\ Min\ y_i$

   9.1. The key goals in both searches are to cross the closest guillotine line (i.e., make the skyline horizontal) and reduce the fitness value by touching the nearest vertex.

      9.1.1. Allowing a box to cut the guillotine line is not permitted, and it is preferred to be in the far-right or far-left of the usable placement line, considering the previous step, 9.1, and explained in Table 5.

10. $w_1^{\#2}.\,Matrix\,(1):\,X_{avi}=90.$

**Table 5.** Sub-matrix after classification step.

| Old | b1 | b2 | b4 | b6 | b7 | b8 | b9 | b10 | b12 | b13 |
|---|---|---|---|---|---|---|---|---|---|---|
| New | b1 | b2 | b3 | b4 | b5 | b6 | b7 | b8 | b9 | b10 |
| $x$i | 70 | 50 | 40 | 40 | 30 | 25 | 25 | 15 | 12 | 10 |
| $y$i | 20 | 30 | 20 | 35 | 15 | 5 | 10 | 10 | 10 | 10 |
| C$_{bi}$ | (35,10) | (25,15) | (20,10) | (20,17.5) | (15,7.5) | (12.5,2.5) | (12,5.5) | (7.5,5) | (6,5) | (5,5) |

    10.1. Reside $b_1$ at $W_{origin}(0.0)$ matrix that belongs to one class divisible for stowing, as shown in (Figure 7).

        10.1.1. Assume $w_1$ is one class generated from a large one, if $w_1$ will subject to classify, then compute the average of $r_i$ and sort it into two classes, one has the objects that have $r_i$ greater than this average, and another has objects that have less than average $r_i$.

    10.2. Extract $b_1$ vertices, $x_{11}^{v1}(0.0),\,x_{11}^{v2}(70.0),\,x_{11}^{v3}(70.20),\,x_{11}^{v4}(0.20).$

    10.3. Select $min\,y_i$ and calculate the next $X_{avi}=90-\left[x_{11}^{v2}(70.0)-x_{11}^{v1}(0.0)\right]=20.$

    10.4. Then, $X_{avi(1-1)}$=20 $\forall x_i \le 20$ [search in $x_i$].

        10.4.1. Create a matrix $w_k$ (1-1) as shown in Table 6.

**Table 6.** The selected boxes suited to be placed.

| X$_i$ branch | b$_8$ | b$_9$ | b$_{10}$ |
|---|---|---|---|
| x$_i \le 20$ | 15 | 12 | 10 |
| y$_i$ | 10 | 10 | 10 |
| C$_{bi}$ | \ | (85.6) | (75.5) |
| r$_i$ | 1.5 | 1.2 | 1 |
| Waste in X | 5 | 8 | 10 |
| Waste in Y | 0 | 0 | ----- |

        10.4.1.1. New $X_{avi(1-1)}=10.$ there are no $x_i \le 10.$ then $\emptyset.$ and $\exists$ in $\forall y_i.$

        10.4.1.2. When zero appears, it means best-fit case and choose $b_{10}.b_9.$

  Then, the placement boxes are $b_1.b_{10}.b_9$ and observe the $y_{10} \le y_9.$

    10.5. Then, $X_{avi(1-1)}$=20 $\forall y_{bi} \le 20$ [search in $y_i$].

        10.5.1. Create a matrix $w_k$ (1-1) as shown in Table 7.

            10.5.1.1. New $X_{avi(1-1)}=5$. Where there are no $y_i \le 5$. Therefore, considered $\emptyset.$ and stop to stow the first trial.

**Table 7.** Sub-matrix excludes products according to X$_{avi.}$

| | b3 | b5 | b6 | b7 | b8 | b9 | b10 |
|---|---|---|---|---|---|---|---|
| xi | 40 | 30 | 25 | 25 | 15 | 12 | 10 |
| $y_i \le 20$ | 20 | 15 | 5 | 10 | 10 | 10 | 10 |
| Cbi | (80,20) | (82.5,15) | (72.5,12.5) | | | | |
| ri | 2 | 2 | 5 | 2.5 | 1.5 | 1.2 | 1 |
| Unused area | 0 | 5 | 15 | 10 | 10 | 10 | 10 |
| | -ve | 0 | ------ | 5 | 5 | 5 | 5 |

10.5.1.2. Select zero waste to achieve the best fit condition. $b_1.b_3$ or $b_1.b_5.b_6$ Branch, then searching in $y_i$ to find two other branches to stow boxes, as shown in Table 8. $\begin{cases} b_1.b_3 \text{......(branch\#y}\rightarrow 1) \\ b_1.b_5.b_{6.(branch\#y\rightarrow 2)} \end{cases}$

10.6. Record the vertices of all free spaces to draw the skyline polygon, as shown in Table 8.

**Table 8.** Select preferred Origin to the Camel or box bi.

|  | b1 | b10 | b9 | b1 | b3 | b1 | b5 | b6 |
|---|---|---|---|---|---|---|---|---|
| $x_{11}^{v1}$ | (0,20) | (--,--) | (80,10) | (0,20) | (70,40) | (0,20) | (70,25) | (75,30) |
| $x_{11}^{v2}$ | (--,--) | (70,10) | (--,--) | (--,--) | (70,20) | (--,--) | (70,20) | (75,20) |
| $x_{11}^{v3}$ | (70,10) | (80,10) | (--,--) | (--,--) | (--,--) | (--,--) | (--,--) | (--,--) |
| $x_{11}^{v4}$ | (70,20) | (--,--) | (80,12) | (70,20) | (--,--) | (70,20) | (--,--) | (90,30) |
| $Min\ x_i, y_i$ | (70, 10) | | (0,20) | | | (0,20) | | |

10.6.1. The selected vortex for branch based on $\exists\ x$ is (70, 10) to be the next origin and determine *Xavi*.

10.6.1.1. Determine $X_{avi(2)} = (80.10) - (70.10) = 10$.

10.6.2. The selected vortex for branch based on $\exists\ y \rightarrow 1$ is (0, 20).

10.6.2.1. Determine $X_{avi(2)} = (70.20) - (0.20) = 70$, as shown in Table 9.

10.6.3. The selected vortex for branch based on $\exists\ y \rightarrow 2$ is (0, 20).

10.6.3.1. Determine $X_{avi(2)} = (70.20) - (0.20) = 70$.

10.6.3.1.1. Select the maximum $X_{avi(2-1)} \leq x_{bi} = 70$, as shown in Table 7, and create the next submatrix $w_{k=2}$, after disposing of all stowed boxes in the selected case above. For this specific branch found $b_1, b_3$ were stowed in a specific location as shown in Table 9.

**Table 9.** Branch and bound for bi at X*avi* <70.

| Left bracnch | b2 | b4 | b5 | b6 | b7 | b8 | b9 | b10 |
|---|---|---|---|---|---|---|---|---|
| $x_i \leq 70$ | 50 | 40 | 30 | 25 | 25 | 15 | 12 | 10 |
| $y_i$ | 30 | 35 | 15 | 5 | 10 | 10 | 10 | 10 |
| $r_i$ | 1.67 | 1.1428 | 2 | 5 | 2.5 | 1.5 | 1.2 | 1 |
| | 20 | 30 | 40 | 45 | 45 | 55 | 58 | 60 |
| Stowage (max), minimum waste area | 10 | 20 | 30 | 35 | 35 | 45 | 48 | |
| | -ve | 8 | 18 | 23 | 23 | 33 | | |
| | -ve | -ve | 3 | 8 | 8 | | | |
| | -ve | -ve | -ve | -ve | stop collecting boxes | | | |
| Right bracnch | b2 | b3 | b4 | b7 | b8 | b9 | b10 | |
| $x_i \leq 70$ | 50 | 40 | 40 | 25 | 15 | 12 | 10 | |
| $y_i$ | 30 | 20 | 35 | 10 | 10 | 10 | 10 | |
| $r_i$ | 1.67 | 2 | 1.1428 | 2.5 | 1.5 | 1.2 | 1 | |
| | 20 | 30 | 30 | 45 | 55 | 58 | 60 | |
| Stowage (max), minimum waste area | 10 | 20 | 20 | 35 | 45 | 48 | | |
| | -ve | 8 | 8 | 23 | 33 | | | |
| | -ve | -ve | -ve | 8 | | | | |
| | -ve | -ve | -ve | 10 | 10 | 10 | 10 | |

| -ve | -ve | -ve | 0 | 0 | 0 | 0 |
|-----|-----|-----|---|---|---|---|
| -ve | -ve | -ve | stop collecting boxes | | | |

10.6.3.1.1.1. $w_{1(1)}\{b_1. b_3\}$ and $w_2\{b_{10}. b_9. b_8. b_7\}$.

10.6.3.2. Select the maximum $X_{avi(2-2)} \leq x_{bi} = 70$ and create the next sub-matrix $w_{k=2}$ after disposing of all stowed boxes in the selected case. Stowed boxes were $b_1. b_5. b_6$, as illustrated in (Figure 8).

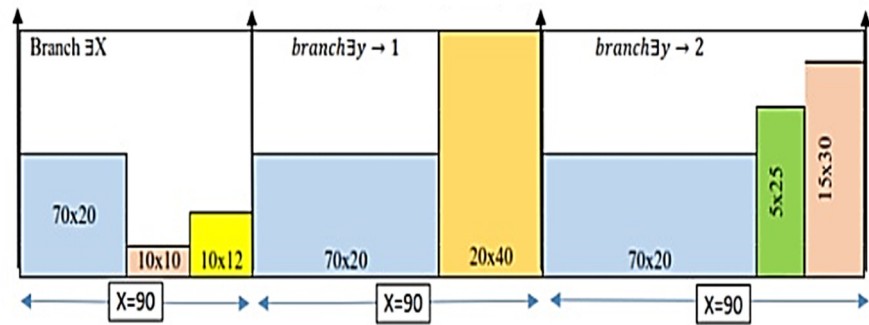

**Figure 8.** Two branches of stowing based on $\vec{X}$ or $\vec{Y}$.

10.6.3.2.1. $w_{k=1(1)}\{b_1. b_5. b_6\}$ and $w_2\{b_{10}. b_9. b_8. b_7\}$.

10.7. When searching about vertices and determine $X_{avi} \leq 5 = \emptyset$, for $\forall \exists x_i | y_i$. Therefore, close this space and raise the vortex to the nearest upper vertex beside the box.

10.7.1. Repeat the previous steps for testing $y_i$.

10.7.1.1. Select the maximum $X_{avi(2-1)} \leq x_{bi} = 20$, as illustrated in Table 10, and following the guillotine line at height 25 then 30 or only for 10 unit-length, and create the next submatrix $w_{k=1}^{\#2}$ after disposal, all stowage/placement boxes were $b_1. b_{10}. b_9$, as shown in Table 11.

**Table 10.** Branch and bound for bi at X*avi* < 20.

|  | b2 | b3 | b4 | b5 | b6 | b7 | b8 |
|---|---|---|---|---|---|---|---|
| $x_i \leq 20$ | 50 | 40 | 40 | 30 | 25 | 25 | 15 |
| $y_i$ | 30 | 20 | 35 | 15 | 5 | 10 | 10 |
| $C_{bi}$ | 20 | 30 | 30 | 40 | 45 | 45 | 55 |
|  | --- | Not meet a condition 9.1 | | | | | |

Place min wastage and then search about minimum fitness nearest guillotine line

**Table 11.** Branch and bound for bi at Y*avi* < 10.

|  | b6 | b7 | b8 |
|---|---|---|---|
| $x_i$ | 25 | 25 | 15 |
| $y_i \leq 10$ | 5 | 10 | 10 |
| $C_{bi}$ | 5 | 2.5 | 1.5 |
|  | 5 | 0 | 0 |
|  | --- | Best fit | |

10.7.1.2. When no $x_i \leq 10 = \emptyset$ found, and try to test the same $X_{avi(2-1)} \leq y_{bi} = 10$, in this case, found $b_8$ and $b_7$ and preferred $b_8$ because $x_8 < x_7$, as shown in (Figures 9 and 10).

10.7.2. Re-record the vertices of all accessible spaces to draw the skyline polygon, as shown in Table 12.

10.7.2.1. Determine $X_{avi(1)} = (90.12) - (80.12) = 10$ and $\exists\forall x_i\emptyset$. whereas when $\exists\forall y_i$ found $w_3^{\#2}$, as illustrated in Table 13. Then, choose $b_7$.

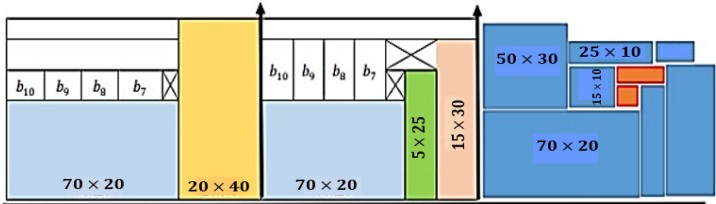

**Figure 9.** Branch of stowing vertically.

**Table 12.** The vertices of all free space.

|  | $B_1$ | $b_8$ | $b_9$ |
|---|---|---|---|
| $x_{11}^{v^1}$ | (0.20) | (70.25) | (80.25) |
| $x_{11}^{v^2}$ | (−.−) | (70.20) | (80.12) |
| $x_{11}^{v^3}$ | (−.−) | (80.12) | (−.−) |
| $x_{11}^{v^4}$ | (70.20) | (80.25) | (90.12) |
| Min $x_i.y_i$ | | (80, 12) | |

**Table 13.** sub-matrix to determine the preferred box bi.

|  | $B_6$ | $b_7$ |
|---|---|---|
| $x_i$ | 25 | 25 |
| $y_i \leq 10$ | 5 | 10 |
| $C_{bi}$ | 5 | 2.5 |
| $r_i$ | 5 | 0 |

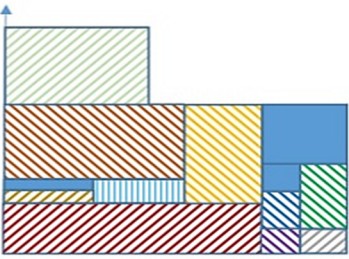

**Figure 10.** Improve the shortest depth branch.

At this level of solution, as illustrated in (Figure 11), the problem is divided into two independent branches, the first based on $x_i$, and the second branch has two branches based on $x_i$ & $y_i$, or allow rotation conditions.

$w_1^{\#2} = \{b_1.b_{10}.b_9.b_8.b_7\}$, and reach $y_{max} = 37$, remain $\{b_2.b_3.b_4.b_5.b_6\}$

$w_{2-1}^{\#2} = \{b_1.b_3.b_{10}.b_9.b_8.b_7\}$, and reach $y_{max} = 40$, remain $\{b_2.b_4.b_5.b_6\}$

$w_{2-2}^{\#2} = \{b_1.b_5.b_6.b_{10}.b_9.b_8.b_7\}$, and reach $y_{max} = 30$, remain $\{b_2.b_3.b_4\}$

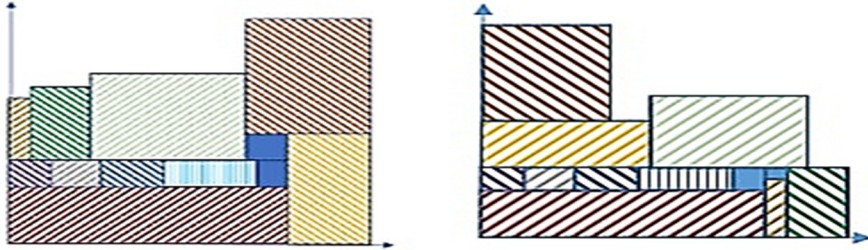

**Figure 11.** Continuo restoring to reduce the waste in stowage area.

All object centroids are considered a vertex of a triangle bound by three conjoined boxes of a replaced object with another that reduces the unused space (i.e., ff the angle $< \beta$ at the centroid vertex of the main item (which not replaced) must be minimum to move the centroid of another inserted object to a horizontal position and $< \beta = 0$. Otherwise, choose inserted objects that reduce $< \beta$ angle. (Figure 4) illustrates the role, when investigating from $\beta_1 < \beta_2$, then an arrangement of (Figure 12A) preferred than (Figure 12B) illustrates that the angle $\beta_1$ is $Cos \, \beta_1 = \frac{a^2 + b^2 - c^2}{2ab}$

When an item is replaced from its potential container $C_{pg}$ it is required to reorder all boxes around this item. The advanced algorithm of replacing an orthogonal item $i$ from container $j$ includes the following steps.

11. Create a new free orthogonal potential container $W_{j'=matrix}$ that replaces objects with similar dimensions equal to the dimensions of the original replaced item, $W_j$. Put into the container $W_{j'}$ a box $b_{i'}$ with the dimensions $w_{i'}^s = w_i^s \, \forall s \in \{1 \dots D\}$, at a point with the coordinates equal to the coordinates of the box $\sum_1^w b_i$ placed into another layer in the same container $W_j$: $x_{i'}^s = x_i^s \, \forall d \in \{1 \dots D\}$.

12. The replacement is subject to two restrictions as follows: The demand based on sequence (classify, select, stow, improve) and the solution effectiveness of the required order $D = \propto p^{-\gamma} + \rho q_v^\delta$ for a long trip, and the customers' requirements rate is variable [33,41,42].

    12.1 $\sum_{i=1}^n b_i \cong X_{avi}$.

    12.2 $min(fitness \, value | \, edges \, length \cong 0. y_{i \cong i+1})$.

        12.2.1 $in \, (wastage \, to \, touch \, the \, guillotine \, line)$.

    12.3 divide this matrix into some subsets considering the following rules:

        $\sum_{i=1}^n b_i \cong X_{avi}$. (To achieve this rule, verify each item's validity before combining it with others, {best-fit} case).

    12.4. $min(fitness \, value | \, edges \, length \cong 0. y_{i \cong i+1})$. This means that a better selection category aligns the heights of the grouped objects with the tallest of them to lead the skyline to be horizontal.

        12.4.1 $min \, (wastage \, to \, touch \, the \, guillotine \, line)$, test $b_1$ with all best-fit of others $b_i$ in matrix $w_k^{\#2}$.

13. It is assumed that objects will permanently be assigned to a specific space and not disposed of, as the customer requirements for these objects always exist in the market in minimum lead-trip. Therefore, the management emphasizes triggering the tour to meet the end-user needs via sequential steps. As cited by M. Schuster et al. [2], autonomation is willing to bear the carrying cost associated with the additional units transporting objects.

14. In point (12) of the OSM-CA, make a new free orthogonal potential container $\boldsymbol{W_{j'=matrix}}$ that contains all of the products subject to the replacement act [38]. Products that are in the container but are not adjacent to the container's edges will be replaced in this example; in other words, they will improve the positions of the objects that belong to the second-class matrix $w_k^{\#2}$, as illustrated in (Figure 12) and Table 14, where Figure 12 illustrates the main two branches (A) and (B) of the final solution deduced from stage 1 implementation.

**Table 14.** The branches solutions obtained from stage-I.

| | X | Y | | | X | Y |
|---|---|---|---|---|---|---|
| $w_1^{\#2}$ | 90 | 77 | Final Solution of Stage 1 | $w_{1\rightarrow branch\_1}^{\#1\&\#2}$ | 98 | 80 |
| $w_2^{\#2}$ | 90 | 90 | | $w_{2\rightarrow branch\_2}^{\#1\&\#2}$ | 98 | 93 |
| $w_3^{\#2}$ | 90 | 80 | | $w_{3\rightarrow branch\_3}^{\#1\&\#2}$ | 98 | 83 |

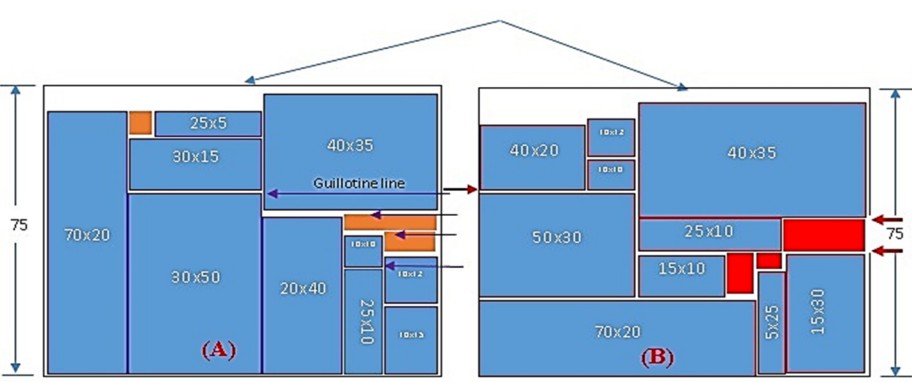

**Figure 12.** Proposed solution for two branches.

*5.2. Calculate the Transportation Cost (Stage-2)*

This section discusses the OSM-CA to solve the stowing problem for each group. The orders (objects) in groups, $1, 2, \ldots, n$. Therefore, the total number of feasible solutions are, $2^n$. The computational effort searching is proportional to, $nA$. Thus, as $n$ increases, the computational effort does not increase exponentially [43, 44], unlike other enumeration techniques that could be used. The OSM-CA presented finding the optimal (least-cost and area) stowing decision as discussed in Algorithm 2.

---

**Algorithm 2. Stage-2: OSM-CA pseudo-codes**.

---

**Begin**

***Step* 1:** Configuration: Set parameters, $T_{\min}$, $T_{\max}$Caravan size (N), the visibility threshold, and initialize each camel's position from Equation (21).

***Step* 2:** Sort the location of each camel using LRV; determine the fitness value of each camel using Equation (1b);

Determine the current best location and fitness in the initial solution.

***Step* 3:** *While* ($iter < number\ of\ trip\ step$) *do*

**do** Number camel caravan size (N)

Compute the temperature of camel ($Tdi, iter$) using Equation (22).

Compute the endurance of camel ($Edi, iter$) using Equation (23)

If $vi, iter$ ($Rand \rightarrow 0$ to 1) < visibility edge, **then**

**Modify** the camel position using Equation (24)

**Otherwise**

**Modify** the Camel position using Equation (21)

End If

*While i*=1:

Convert position camel to travel sequence using LRV and determine the total distribution cost in each camel. If fitness, the new positions are preferred. The new best is the global best and saves the best solution (fitness and location)

End If

**Assign** new visibility for each camel

*Step* **4:** End While

*Step* **5:** Output the best solution

**End**

Let $(j, x)$ be the lowest cost of any stowing decision considering the first $j$ (objects) and required area $x$ on the last unplanned drawer.

**Configuration:** $f(0; 0) = 0$. $f(0; A) = S. f(0; x) = infinity$ if $0 < x < A$.

**Recursion:** If $x = 0$, then $f(j, x) = f(j - 1, x) + F_j$, in this case, no orders are stowage. While if $0 < x \leq e_i$ then $f(j, x) = min\{f(j - 1, x) + F_j, f(j - 1, A + x - e_j) + (s_j + 1)R\}$. In this situation, also order $j$ had not been stowed, or order $j$ added its $s_j$ (returns) to the drawer and needs an extra area $e_j$ for order $j$ that fit onto the last returns to the stowage. In this case, the portion of order $j$ on the new area of the returns drawer has an area $x$, and the portion on the last returns drawer that has an area $e_j - x$. Therefore, the area on the last assigning drawer for these returns describes as $A - (e_j - x) = A + x - e_j$ before adding order $j$ to the stowage map. If $e_j < x \leq A$, then $f(j, x) = min\{f(j - 1, x) + F_j, f(j - 1, A + x - e_j) + (s_j)R\}$. In this case, either order $j$ was not stowed, or order $j$ added its $s_j$, but the extra area of order $j$ fits onto the last unplanned area for the returns objects. This step means $\min_{0 \leq x \leq A}\{f(n, x)\}$, the complexity of tackling this problem is $O(nA)$. This preparation allows for identifying a case that can be solved immediately. Note that $(s_j + 1)R$ is an upper bound for the cost of stowing order, $j$. If $(s_j + 1)R \leq F_j$, then the cost of adding the order to the nest is less than or equal to the cost of excluding it. If $(s_j + 1)R \leq F_j \forall j$, then stowing no orders is the only solution that could cost less than stowing every order. To decide, compare $\sum_j F_j$ (the cost of stowing no orders) to, $S + [\sum_j O_j / A]R$, which means (the cost of stowing every order). If the first quantity is smaller, stowing no orders is the optimal solution. Otherwise, stowing every order is an optimal solution [45].

15. *Linear Programming Reduction:* We build and solve a linear programming relaxation of the dynamic stowing problem to construct an efficient heuristic. The variables have been changed to be continuous, and the objective function has been changed to be linear.

**Minimize:**

$$C^R(X) = \sum_{j \in G} F_j + \sum_{j \in G} (s_j R + \frac{e_j}{A} R - F_j) X_j + YS + \sum_{i=0}^{L} \sum_{j=0}^{L} \sum_{k=1}^{K} Cf.LPK.d_{ij}.x_{ijk} + \sum_{j=1}^{L} (max(0, (S_j - LT_j))).LCT \quad (35)$$

Can define the ideal stowage according to: Let $T_j = (s_j R + \frac{e_j}{A} R - F_j)$ be the whole cost of stowing orders, $J_j$ as expressed in Eq. (35). If $T_j < 0$, indicate non-stowing case (all $X_j < 0$ *and* $Y = 0$). Otherwise, some $T_j < 0$. Let $w^{\emptyset} = \{j: T_j < 0\}$. If $\sum_{j \in W}(T_j + S) \geq 0$, indicates a non-stowing situation also. While if $\sum_{j \in W}(T_j + S) < 0$, then the optimal solution is to stow order $J_j$, when $j \in w(X_j = 1)$ and if $j \in w(Y = 1)$.

1. Calculate the estimated total cost $T_j$ of stowing order $J_j$ for all $j = 1, \dots, n$, in group $b_G$:

2. $T_j = (s_j R + \frac{e_j}{A} R - F_j)$.

3. *Let* $w^{branch} = \{j: T_j < 0\}$.

4. *If* $w$ is empty $= \emptyset$ or $\sum_{j \in W}(T_j + S) \geq 0$, then stow nothing. Otherwise, since $\sum_{j \in W}(T_j + S) < 0$, stow order $J_j(X_j = 1)$ if and only if $j \in W$.

If the $X^0$ is the stowage map that the heuristic creates and $X^*$ is an optimal stowage for the OSM-CA, then $C(X^0) - C(X^*) < R$. While for any solution $X, C(X)$ is the cost of the solution, where discussed by Eq. (36)

$$C(X) \geq C^R(X) = \sum_{j \in G} F_j + \sum_{j \in G} T_j X_j + YS \quad (36)$$

Transportation managers must first arrange the orders needed during each timespan into clusters, relying on their form and size to use dynamic stowing OSM-CA, via the following indicators expressed in Eqs. (35, 36).

- $F_j$, indicates the whole prep and penalty cost of each order, $J_j$, whether it is not stowed and/or transported.

- $O_j$, the whole required area to stow order, $J_j$.

- $A$, the serviceable area of a returns items.

- $R$, the labor and drawer loading cost of returns objects.

- $S$, the vehicle prep cost for a stowage.

Aimed at any order, $J_j$, calculating, $F_j$, via determining the planned packing items; $N(X)$, and calculate $O_j$, multiple the parts inside the order by the needed area per part as expressed in Eq. (37), (including useless and/or interpreted spacing) [46, 47]. The data have been aggregated dynamically via IoT to feed the proposed algorithm. Therefore,

$$T_j = \left(\frac{O_j}{A} R - F_j\right) \quad (37)$$

## 6. Results and Discussion

With the advancement of sophisticated and digital manufacturing technology, it is now possible to completely utilize circular resources. However, there is currently a dearth of study that examines how digital technology may affect the development of a circular economy in the context of supply chains [48, 49]. This study aims to evaluate the impact of technological innovation on CE practices and their connection to economic and environmental performance, as discussed by Khan, S. et al. (2022) [50,51]. Only in China can

economic expansion and frequent travel play such a big role in pushing up crude oil imports. Foreign direct investment and industrialization both play a significant influence [52]. As a result, the suggested methodology is based on maximizing the volume of items transported while simultaneously decreasing the number of transportation trips. Consequently, cut back on fuel usage. There are three differences between the proposed methodology and the classic placement approaches for strip packing or others. The first is the limited container size (width, length, and height), which represents the standard size. The search for customizing specific objects using a modified camel algorithm aided by the mathematical procedures interested in the parameters shown in Table 2 to fit best the inner of the container. The modified metaheuristic can be programmed using C-Sharp software to create the visual map for the stowage object inner in this fixed area, which is discussed in Section 5.2. Finally, track the map (loading and unloading and loading returned objects from different stations along the trip path), which form the main framework discussed in Figure 2. The branch-and-bound methodology [4, 47, 53, 54] may be classic in the implementation of small-size problems. At the same time, customizing many objects is considered to be an NP-hard problem and needs an innovative methodology to help achieve the e-commerce target discussed before. In 2020–2021, the sudden (COVID-19), the "black swan" outbreak, struck every country, sparking a global pandemic, which led to badly disrupted logistic activities for manufacturing companies, and retailer stores, consequently increasing the delivery lead time and customer grumbles [55]. Therefore, the proposed methodology interested in increasing the number of transported items to reduce the delivery lead time and partially satisfy customers. E-commerce data are characterized by quick growth [56], which pushes the researchers to create modern analysis models to simplify the management of these data means using IoT and sustainable supply chain management (SSCM) [57], which refreshes the circular economy, also referred to as "Blockchain" technology, as a distributed digital ledger technology, guarantees security, visibility, traceability, and transparency and promises ease for environmental and global supply chain problems. Retailer distribution is an example of a circular economy that must be sustainable [58, 59]. This concept is compatible with the proposed, which works on accelerating the distribution cycling of e-commerce. Some researchers identified and faced some influential barriers to using the IoT and analyzed the causes of weak supply chain sustainability [60]. Therefore, the authors have interested in tackling the problem from its root by trying to increase transported items and accelerate the circular economy and enhance the e-commerce processes. Table 15 shows the tuned parameters for ACO, Camel algorithm, and dynamic OSM-CA. Both ACO, CA, and proposed heuristics that the numerical model supported are compared with LINGO exact output (Mathematical Model). Table 16 shows the ranges for the assumed cost of moving the actuators to their original location and the cost of replacement achieved.

**Table 15.** Tuned algorithm parameters.

| | $\alpha$ | $\gamma$ | $Q$ | $\tau_0$ | $\rho$ | $N$ | $IT$ |
|---|---|---|---|---|---|---|---|
| **ACO** | 1 | 1 | 5000 | 0.1 | 0.83 | 20 | 500 |
| | $T_{min}$ | $T_{max}$ | ------ | visibility | ------ | Caravan size (N) | $IT$ |
| **CA** | 5 | 200 | Ign. | 0.05 | 0.80 | 20 | 500 |
| **OSM-CA** | 2 | 200 | Ign. | 0.03 | 0.81 | 20 | 500 |

**Table 16.** The range of transported costs per unit area.

| Range of back to the original location cost | Scope of variable cost in 1st case (demand) | Scope of variable cost in 2nd case (returns) |
|---|---|---|
| | All costs mentioned in the context | |
| (30–50) * Avg. replacing cost | 10–30 | 10–50 |

Figure 13 illustrates the percentage of deviation for the four data sets identified (Table 1) about the optimal solution extracted by LINGO (mathematical model) and shows the superiority of the proposed methodology over the native camel and ant-colony algorithms by 0.417% and 2.0528%. While if the problem size is over 500 objects, the LINGO consumes significant time, exceeds 24 h, and often fails to extract the solution, as illustrated in (Figure 14). The results deviations are represented and summarized in Tables 17–19 in Relative Percentage Deviation (*RPD*). The total cost and related area are also shown in the same tables, where *RPD* is calculated as in Equation (38).

$$RPD_{i(ACO,SA,ESMM)} = \frac{OSMCA - LINGO}{LINGO} \times 100 \; \forall i = 1,2,3 \qquad (38)$$

The computational analysis shows that OSM-CA provides better results than ACO or CA solo. The proposed methodology provides significantly better solutions with a difference of about 3.045% for ACO, 1.75% for CA, and 1.47% for OSM-CA from the optimal solutions. The *p*-values are calculated for the average deviations between ACO and OSM-CA to be 0.0056, which is <0.05.

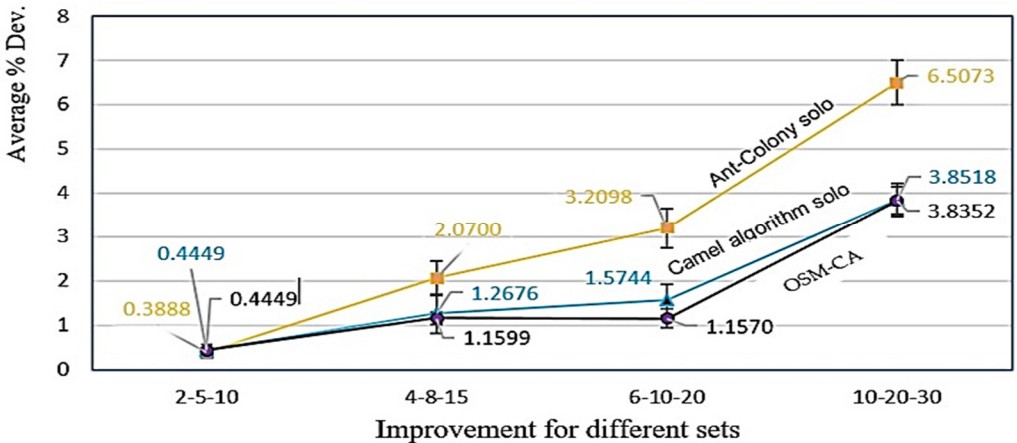

**Figure 13.** The deviation of proposed methodology OSM-CA and ACO, CA methods with Mathematical equation effect result.

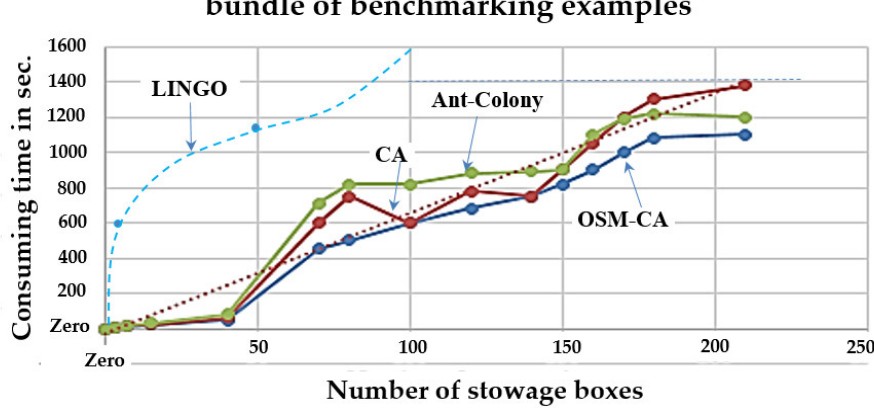

**Figure 14.** The time required to create a stowage map for each drawer (potential container).

*Analysis of Results*

Excellent e-commerce needs a quick decision for loading the demands and/or the returned objects during the transportation path, which may cause the arrival time among

the stations to be less than 20 min. As the number of instances and/or the boxes grow to more than 100 that must be stowed at the same time, the mathematical models cannot find an exact solution via LINGO software for running over 9 h to present a stowage solution in the available space that appears after unloading the requested objects at clients' stations. Therefore, the authors developed one of the optimization algorithms (i.e., the camel algorithm) by embedding mathematical equations to make oriented stowage and programming it with the C-sharp software to present a visual solution named alluded "stowage map", which is named (OSM-CA) using a Laptop that has processor Core i5 and 16G Ram. This map helps increase the stowing capacity for the transported products, reducing the total transportation costs, the number of trips, and the waiting time at each station (i.e., Lean objectives). The Lean-IoT achieves many of the lean objectives and provides evidence of the effectiveness of the proposed OSM-CA methodology, as illustrated in Figures 13 and 14 and shown in Tables 17-19. On average, the proposed algorithm gives a much better solution than the precise solution, with a difference of roughly 1.47% for OSM-CA. The findings of the computational study demonstrate that OSM-CA is cumulative superior to the ant-colony and CA solo by 12% and 5%, respectively, as shown in (Figure 13). Appendix-A explains the comparability between the proposed algorithm and the other two algorithms, such as the ant-colony and the camel algorithm.

**Table 17.** Results obtained using LINGO, ACO, CA, and OSM-CA for group 1 that distributed 10 different boxes area in a container dimension $80_{\bar{x}} \times 90_{\bar{y}} = 720{,}000 \, mm^2$.

| Prob. Inst. | Total Unused Area $mm^2$ | | | | Unused Area Deviation % | | | Prob. Inst. | Total Unused Area $mm^2$ | | | | Unused Area Deviation % | | |
|---|---|---|---|---|---|---|---|---|---|---|---|---|---|---|---|
| | LINGO | ACO | CA | OSM-CA | ACO | CA | OSM-CA | | LINGO | ACO | CA | OSM-CA | ACO | CA | OSM-CA |
| 1 | 5190 | 6900 | 6750 | 6295 | 0.32% | 0.30% | 0.21% | 16 | 13322 | 13411 | 13407 | 13401 | 0.67% | 0.64% | 0.6% |
| 2 | 11497 | 11511 | 11499 | 11491 | 0.12% | 0.02% | −0.1% | 17 | 15790 | 15816 | 15790 | 15790 | 0.16% | 0.00% | 0.0% |
| 3 | 15037 | 15090 | 15064 | 15061 | 0.35% | 0.18% | 0.2% | 18 | 16894 | 16927 | 16894 | 16891 | 0.20% | 0.00% | 0.0% |
| 4 | 13097 | 13130 | 13099 | 13092 | 0.25% | 0.02% | 0.0% | 19 | 15565 | 15582 | 15565 | 15561 | 0.11% | 0.00% | 0.0% |
| 5 | 11981 | 12199 | 12309 | 12303 | 1.82% | 2.74% | 2.7% | 20 | 16971 | 17077 | 17028 | 17022 | 0.62% | 0.34% | 0.3% |
| 6 | 11477 | 11560 | 11500 | 11500 | 0.72% | 0.20% | 0.2% | 21 | 14891 | 14982 | 14961 | 14961 | 0.61% | 0.47% | 0.5% |
| 7 | 15612 | 15887 | 15839 | 15832 | 1.76% | 1.45% | 1.4% | 22 | 13728 | 13736 | 13728 | 13721 | 0.06% | 0.00% | −0.1% |
| 8 | 15573 | 15780 | 15843 | 15841 | 1.33% | 1.73% | 1.7% | 23 | 14054 | 14068 | 14054 | 14052 | 0.10% | 0.00% | 0.0% |
| 9 | 14221 | 14249 | 14221 | 14220 | 0.20% | 0.00% | 0.0% | 24 | 15784 | 16077 | 16035 | 16031 | 1.86% | 1.59% | 1.6% |
| 10 | 12785 | 12799 | 12785 | 12781 | 0.11% | 0.00% | 0.0% | 25 | 13719 | 13775 | 13735 | 13731 | 0.41% | 0.12% | 0.1% |
| 11 | 12138 | 12281 | 12211 | 12210 | 1.18% | 0.60% | 0.6% | 26 | 12897 | 12942 | 12904 | 12900 | 0.35% | 0.05% | 0.0% |
| 12 | 13769 | 13992 | 13961 | 13961 | 1.62% | 1.39% | 1.4% | 27 | 13534 | 13578 | 13536 | 13531 | 0.33% | 0.01% | 0.0% |
| 13 | 13824 | 13843 | 13824 | 13821 | 0.14% | 0.00% | 0.0% | 28 | 10117 | 10192 | 10125 | 10121 | 0.74% | 0.08% | 0.0% |
| 14 | 14934 | 15062 | 15008 | 15001 | 0.86% | 0.50% | 0.4% | 29 | 14377 | 14438 | 14424 | 14421 | 0.42% | 0.33% | 0.3% |
| 15 | 11808 | 11879 | 11870 | 11870 | 0.60% | 0.53% | 0.5% | 30 | 14253 | 14263 | 14253 | 14251 | 0.07% | 0.00% | 0.0% |
| The average deviation | | | | | | | | | | | | | 0.61% | 0.45% | 0.42% |

**Table 18.** The comparison among different groups for an unused area $ft^2$.

| | LINGO Time Window | ACO | Time Window | CA | Time Window | OSM-CA | Time Window |
|---|---|---|---|---|---|---|---|
| **10 boxes** | 3 min | 0.3888% | 6 s | 0.4449% | 6 s | 0.4449% | 6 s |
| **50 boxes** | 67 min | 2.070% | 13.2 s | 1.2676% | 17 s | 1.1599% | 17 s |
| **100 boxes** | 490 min | 3.2098% | 75 s | 1.5744% | 38 s | 1.5570% | 38 s |
| **500 boxes** | 1440 min | 6.5073% | 108 s | 3.8518% | 90 s | 3.8352% | 60 s |
| | | 3.045% | | 1.75% | | 1.47% | |

**Table 19.** The expected transportation costs obtained using LINGO and OSM-CA [4].

| | The $b_{G1}$ Instances Results | | | | The $b_{G2}$ Instances Results | | | | The $b_{G3}$ Instances Results | | | | The $b_{G4}$ Instances Results | | |
|---|---|---|---|---|---|---|---|---|---|---|---|---|---|---|---|
| Prob. Inst. | LINGO | OSM-CA | Dev.% | Prob. Inst. | LINGO | OSM-CA | Dev. | Prob. Inst. | LINGO | OSM-CA | Dev. | Prob. Inst. | LINGO | OSM-CA | Dev. |
| 1 | 128040 | 128524 | 0.38% | 1 | 177854 | 178112 | 0.15% | 1 | 236721 | 237741 | 0.43% | 1 | 314390 | 323986 | 3.05% |
| 2 | 114971 | 114997 | 0.02% | 2 | 209060 | 209832 | 0.37% | 2 | 228294 | 228752 | 0.20% | 2 | 280884 | 284218 | 1.19% |
| 3 | 150375 | 150646 | 0.18% | 3 | 167758 | 167990 | 0.14% | 3 | 210147 | 211125 | 0.47% | 3 | 335035 | 354308 | 5.75% |
| 4 | 130971 | 130997 | 0.02% | 4 | 179626 | 179906 | 0.16% | 4 | 228458 | 229740 | 0.56% | 4 | 312855 | 325247 | 3.96% |
| 5 | 119819 | 123098 | 2.74% | 5 | 202973 | 203677 | 0.35% | 5 | 247414 | 248696 | 0.52% | 5 | 286988 | 298794 | 4.11% |
| 6 | 114775 | 115001 | 0.20% | 6 | 166785 | 168870 | 1.25% | 6 | 219715 | 223004 | 1.50% | 6 | 324858 | 332985 | 2.50% |
| 7 | 156127 | 158396 | 1.45% | 7 | 182584 | 185930 | 1.83% | 7 | 216806 | 220383 | 1.65% | 7 | 315778 | 328988 | 4.18% |
| 8 | 155732 | 158438 | 1.74% | 8 | 154217 | 154532 | 0.20% | 8 | 230437 | 230437 | 0.00% | 8 | 307531 | 317144 | 3.13% |
| 9 | 142211 | 142211 | 0.00% | 9 | 179461 | 180010 | 0.31% | 9 | 225189 | 225857 | 0.30% | 9 | 342149 | 355502 | 3.90% |
| 10 | 127857 | 127857 | 0.00% | 10 | 173401 | 175910 | 1.45% | 10 | 238507 | 239051 | 0.23% | 10 | 355439 | 365249 | 2.76% |
| 11 | 121383 | 122113 | 0.60% | 11 | 195559 | 197057 | 0.77% | 11 | 227820 | 229522 | 0.75% | 11 | 321067 | 338172 | 5.33% |
| 12 | 137696 | 139617 | 1.40% | 12 | 185327 | 185890 | 0.30% | 12 | 212515 | 214867 | 1.11% | 12 | 356106 | 368258 | 3.41% |
| 13 | 138248 | 138248 | 0.00% | 13 | 171036 | 174245 | 1.88% | 13 | 242507 | 246847 | 1.79% | 13 | 326465 | 344397 | 5.49% |
| 14 | 149346 | 150085 | 0.49% | 14 | 203715 | 204715 | 0.49% | 14 | 276236 | 280983 | 1.72% | 14 | 345003 | 354941 | 2.88% |
| 15 | 118089 | 118701 | 0.52% | 15 | 156007 | 161107 | 3.27% | 15 | 245646 | 248857 | 1.31% | 15 | 303481 | 308457 | 1.64% |
| 16 | 133229 | 134079 | 0.64% | 16 | 193382 | 193690 | 0.16% | 16 | 231457 | 232862 | 0.61% | 16 | 326392 | 332121 | 1.76% |
| 17 | 157904 | 157904 | 0.00% | 17 | 206042 | 206837 | 0.39% | 17 | 238796 | 238930 | 0.06% | 17 | 350396 | 359350 | 2.56% |
| 18 | 168949 | 168949 | 0.00% | 18 | 214247 | 215589 | 0.63% | 18 | 258653 | 258975 | 0.12% | 18 | 314917 | 342152 | 8.65% |
| 19 | 155656 | 155656 | 0.00% | 19 | 224956 | 228345 | 1.51% | 19 | 229847 | 235665 | 2.53% | 19 | 284996 | 300637 | 5.49% |
| 20 | 169718 | 170284 | 0.33% | 20 | 178472 | 178702 | 0.13% | 20 | 278730 | 281972 | 1.16% | 20 | 305625 | 317187 | 3.78% |
| 21 | 148915 | 149618 | 0.47% | 21 | 197683 | 200418 | 1.38% | 21 | 258885 | 259288 | 0.16% | 21 | 338830 | 348824 | 2.95% |
| 22 | 137285 | 137285 | 0.00% | 22 | 186321 | 186753 | 0.23% | 22 | 224956 | 236313 | 5.05% | 22 | 319456 | 339354 | 6.23% |
| 23 | 140544 | 140544 | 0.00% | 23 | 211361 | 215005 | 1.72% | 23 | 214213 | 217430 | 1.50% | 23 | 288152 | 301301 | 4.56% |
| 24 | 157845 | 160355 | 1.59% | 24 | 166220 | 166220 | 0.00% | 24 | 244412 | 251303 | 2.82% | 24 | 322640 | 336138 | 4.18% |
| 25 | 137194 | 137353 | 0.12% | 25 | 191598 | 191976 | 0.20% | 25 | 227719 | 230432 | 1.19% | 25 | 342406 | 365571 | 6.77% |
| 26 | 128974 | 129044 | 0.05% | 26 | 232565 | 233243 | 0.29% | 26 | 250476 | 261000 | 4.20% | 26 | 335441 | 340769 | 1.59% |
| 27 | 135349 | 135362 | 0.01% | 27 | 161654 | 162481 | 0.51% | 27 | 227589 | 228077 | 0.21% | 27 | 333252 | 341876 | 2.59% |
| 28 | 101177 | 101255 | 0.08% | 28 | 191584 | 193258 | 0.87% | 28 | 240589 | 242381 | 0.74% | 28 | 310810 | 322600 | 3.79% |
| 29 | 143779 | 144241 | 0.32% | 29 | 156107 | 158644 | 1.63% | 29 | 202918 | 205225 | 1.14% | 29 | 324880 | 331498 | 2.04% |
| 30 | 142535 | 142535 | 0.00% | 30 | 161654 | 181459 | 12.25% | 30 | 229604 | 231205 | 0.70% | 30 | 328657 | 344555 | 4.84% |
| The average deviation | | | 0.67% | The average deviation | | | 2.23% | The average deviation | | | 1.19% | The average deviation | | | 1.7% |

Table 19 indicates the efficiency of the proposed OSM-CA for cost indicators. The authors observe that the deviation for Group 2 is unpredictable, where the expected was lower than Group 3, but have no answer except them devoid of biased results.

### 7. Conclusions

This work has been adopted by several famous e-commerce companies in KSA and Egypt. The proposed algorithm "OSM-CA" justifies and explains the metaheuristics development to tackle large problems in minimum time and can handle them via the IoT [61]. The stowage map extracted more rapidly than LINGO by 60% at stowing 100 objects have different sizes, while superior to the ant colony and CA over all 400 stowed objects by 27% and 9%, respectively. The OSM-CA achieved a 28% reduction in the waiting time at various stations, reduced the over-processing activities by 35%, which accelerated the logistic trip (i.e., reflected on the total delivery time), and reduced the holding and transportation costs by the same amount, as shown in Table 20. On the sidelines of the article, if the number of trips is reduced due to increasing in stowage capacity through transportation, the fuel consumption is reduced [24, 49, 61]. The main innovation of this work is extracting a visual stowage map guide for the driver for customizing the objects in an assigned area that achieves minimum waste in the stowage areas or unused spaces, as expounded in Appendix A.

**Table 20.** The KPI of superior the proposed algorithm OSM-CA-IoT, according to the e-commerce objectives for more than 120 randomly examples.

| Verification of Superiority of the KPI's OSM-CA | Ant-Colony | CA |
| --- | --- | --- |
| Maximum depth exploited at ≤100 objects | 25% | same |
| Maximum depth exploited at ≥101 objects | 9% | 21% |
| Unused Spaces [Table 17] | 19% | 3% |
| Extracting the stowage solutions [(Table 18] | 1.6% | 0.28% |
| Over-processing | 35% | 35% |
| Waiting time reduced by | 28% | 14% |
| Delivery time reduced by | 14% | 14% |
| The average number of trips reduction | 17.5% | 2.76% |
| Transportation costs decreased | 6.9% | 5.96% |
| The total efficiency of solution < 40 boxes (Figure A2) | 15.8% | 30% |
| The total efficiency of solution > 40 boxes (Figure A2) | 27% | 17.2% |
| Customer satisfaction, according to VOC | Up 8% | Up 8% |

In Appendix A (Figure A1) illustrates the data inputs for the returned objects at a specific station aggregated via IoT to be stowed in preferred spaces that achieve the article's objectives. Those who adopt the proposed methodology enhance their e-commerce for specific products of many different sizes (i.e., bathtubs). They manage their transportation system via IoT for seven stations (i.e., aggregate the clients' demands, extract the stowage map to indicate preferred loading locations, exchange the returned products until they reach the next station by minute, and select the oriented products). They emphasize the superiority of OSM-CA over the competitors discussed in this article by quickly creating the visual stowage map, which is considered a helpful guide for the stowage process. Figure A2 illustrates the synopsis of the aggregated data of the objects that must be loaded and unloaded at different stations on the transportation path for one of the most famous companies in Egypt, which is based on data derived from Tables 17–19. Figure A2 illustrates that the mathematical model loses its efficiency in finding a preferred solution with an increase in the number of products that must be stowed in over 200 boxes. Figure A3 discusses if the safety boundary between the stowed boxes is urgent or may be neglected. Figures A4 and A5 illustrate the excellence of the ant-colony, which stows nine objects out of ten over the camel algorithm, which stows seven objects only, while the proposed stows all ten objects, as illustrated in (Figure A6) and (Figure A7). While if the number of objects transported increases over 400, the camel algorithm presents a preferred stowage map, reducing the unused area in the container. Figures A8–A13 discuss the contribution of OSM-CA in decreasing the unused stowed area in the containers over

their competition, as shown in Table 19. The authors implement the stowing in the potential containers called drawers as discussed by Ahmed M. Abed et al. (2022) [48,53] with two sizes: 80 × 90 and 160 × 180 inches.

## 8. The Future Work

The future work for this research point is developing the IoT to classify the products by using the harmony search algorithm according to profitability before creating the stowage map at each station and redesigning the best drawer sizes and whole trucks according to the companies' needs via developing the digital twins' program.

**Author Contributions:** Conceptualization, A.M.A. and L.F.S.; methodology, A.M.A.; software, A.M.A.; validation, A.M.A., and L.F.S.; formal analysis, A.M.A.; investigation, A.M.A.; resources, A.M.A.; data curation, A.M.A.; writing—original draft preparation, A.M.A.; writing—review and editing, A.M.A.; visualization, A.M.A.; supervision, A.M.A. and L.F.S.; project administration, A.M.A.; funding acquisition, A.M.A and L.F.S. All authors have read and agreed to the published version of the manuscript.

**Funding:** The authors extend their appreciation to Deputyship for Research and Innovation, Ministry of Education in Saudi Arabia, for funding this research work through the project number (IF-PSAU-2021/01/19104).

**Data Availability Statement:** https://drive.google.com/drive/folders/1TUjqXVQqkjWSvJwLx3WZ8g9lsG3udXAc?usp=sharing.

**Conflicts of Interest:** The authors declare no conflict of interest.

## Appendix A

**Figure A1.** Data of the return products of first station on distribution path aggregated via IoT.

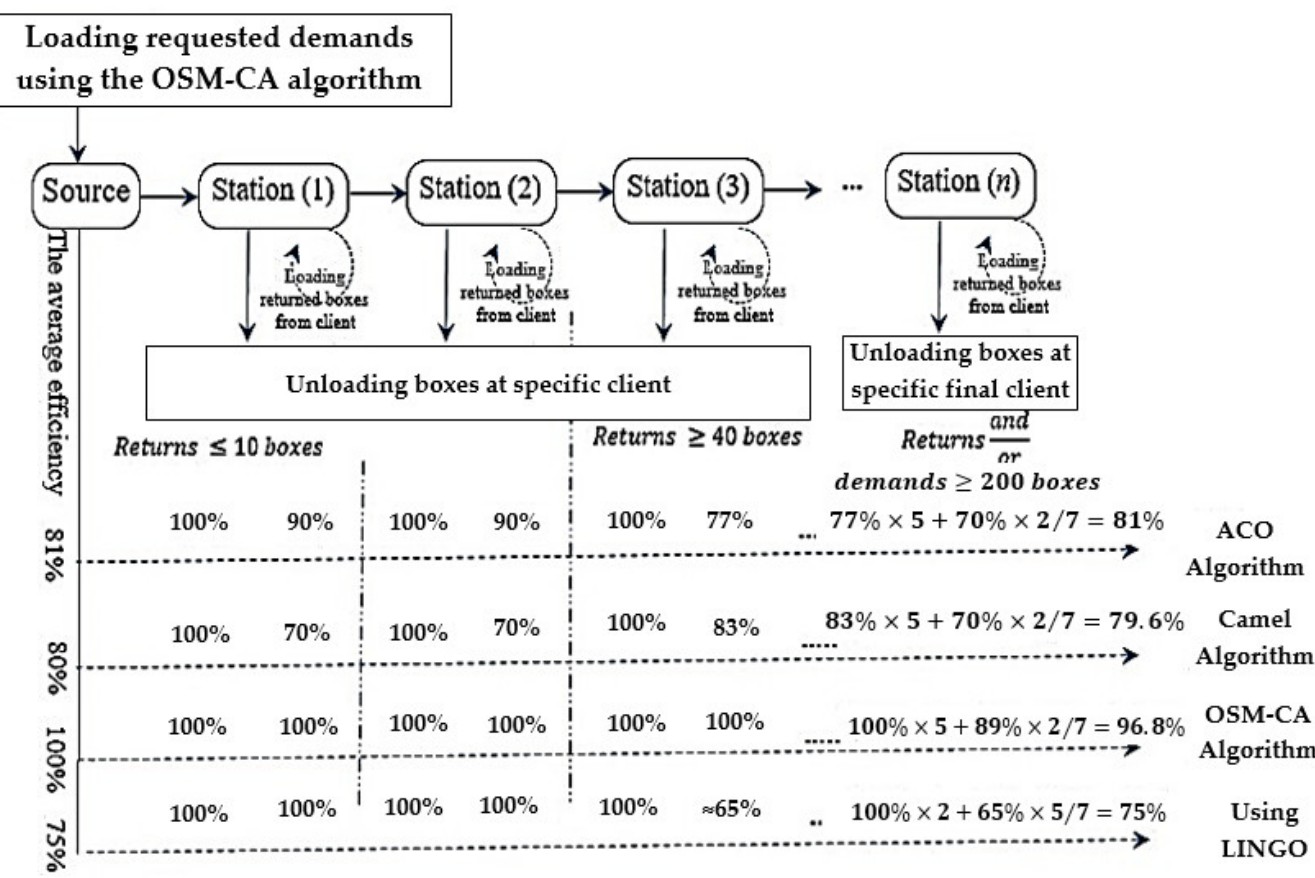

**Figure A2.** The efficiency of transportation system on distribution path aggregated via IoT for seven stations (n = 7).

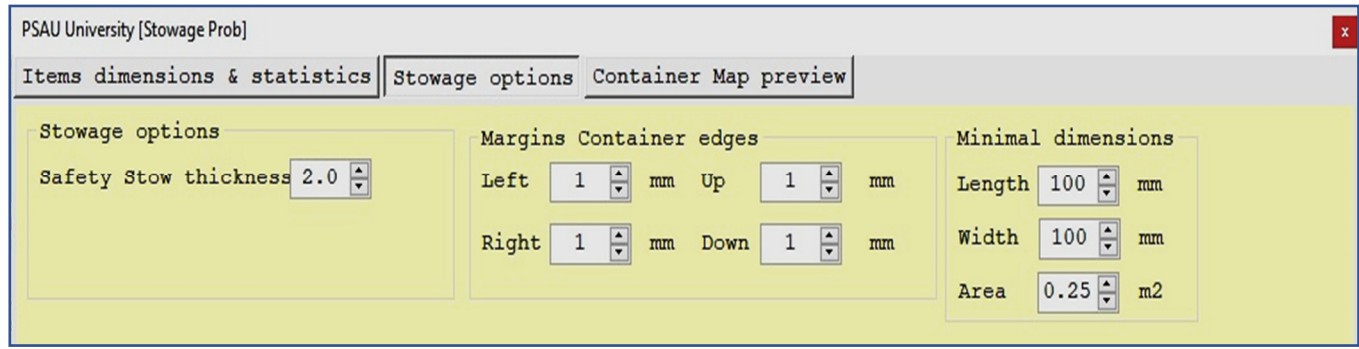

**Figure A3.** Safety distance of the return products when stowage [0: < Y].

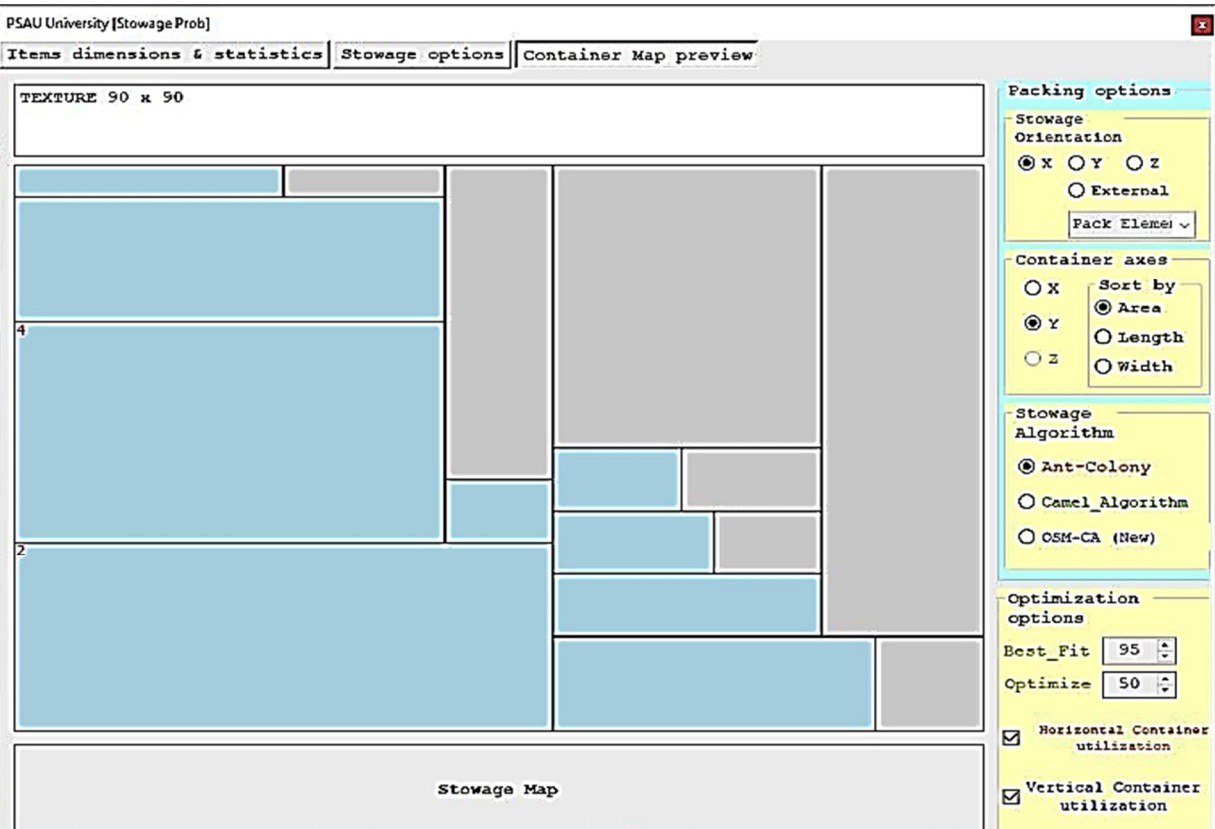

**Figure A4.** The stowage map for nine objects by the ant-colony algorithm optimization for first station on the distribution path.

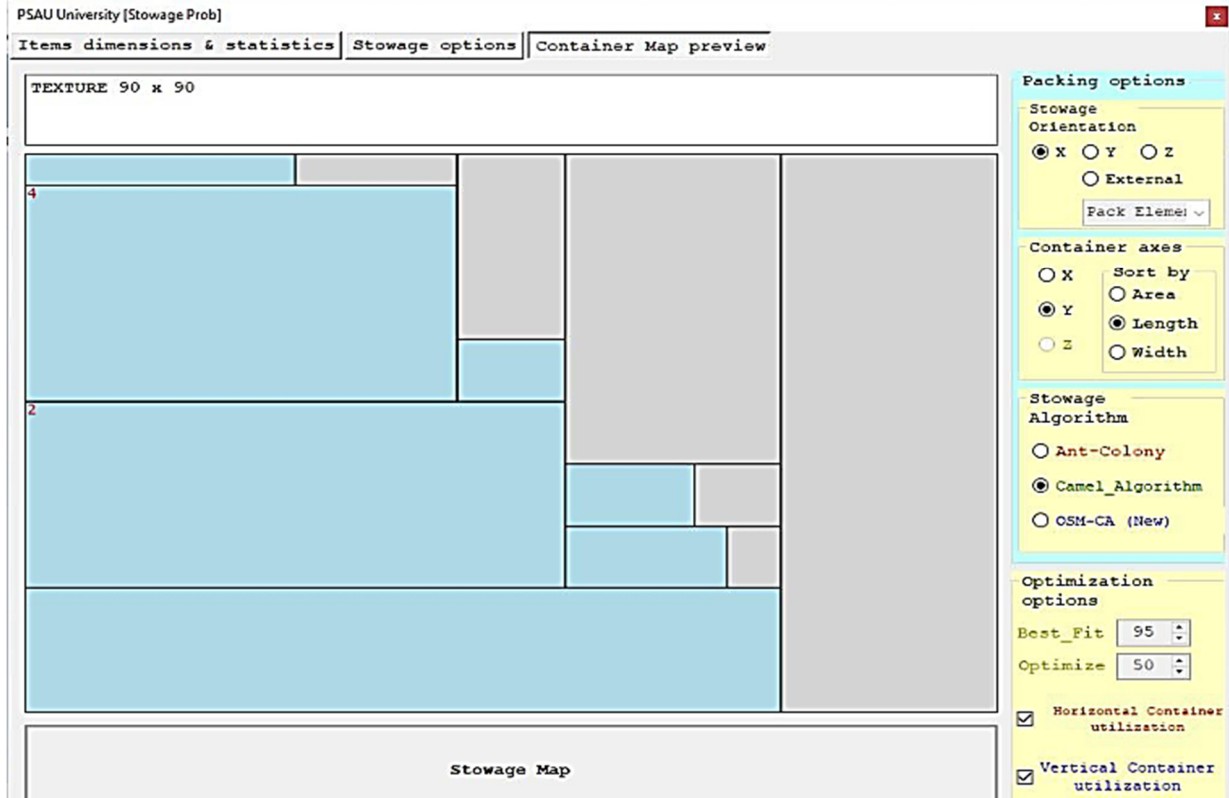

**Figure A5.** The stowage map for seven objects by the camel algorithm optimization for first station on the distribution path.

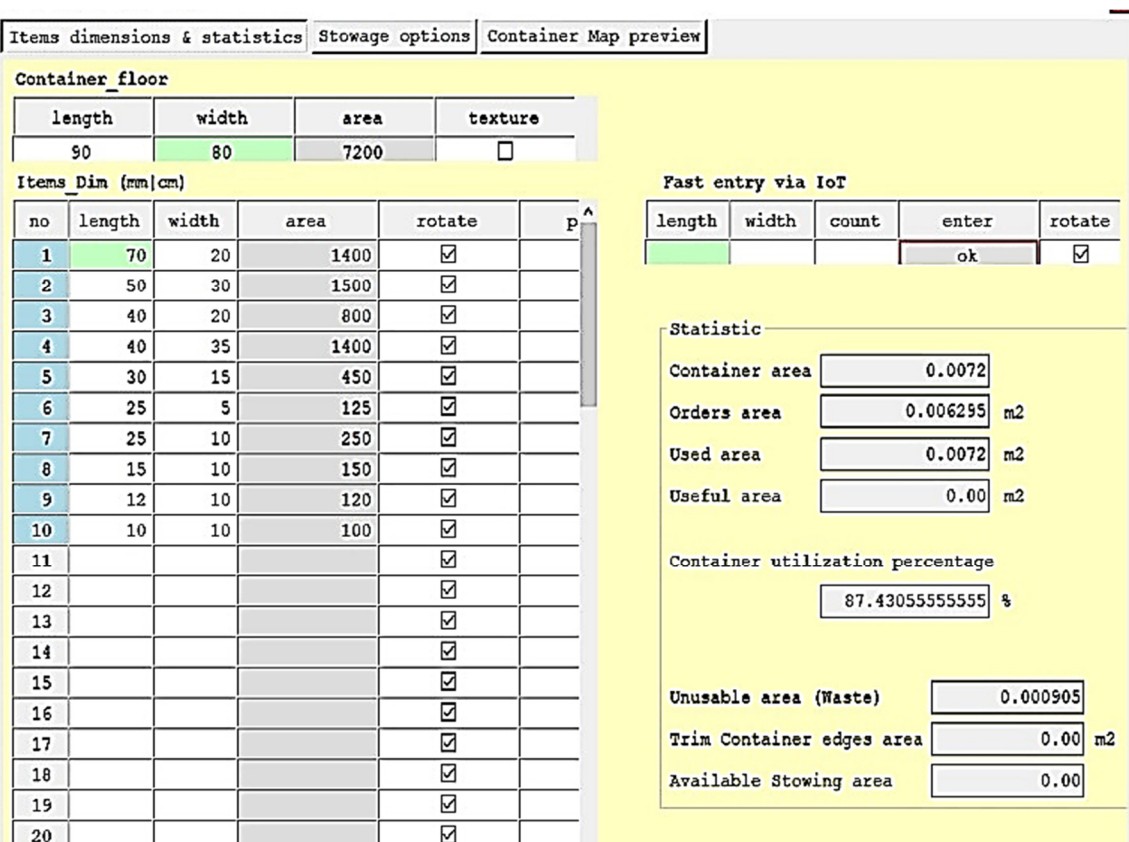

**Figure A6.** The container utilization by OSM-CA for first station on the distribution path.

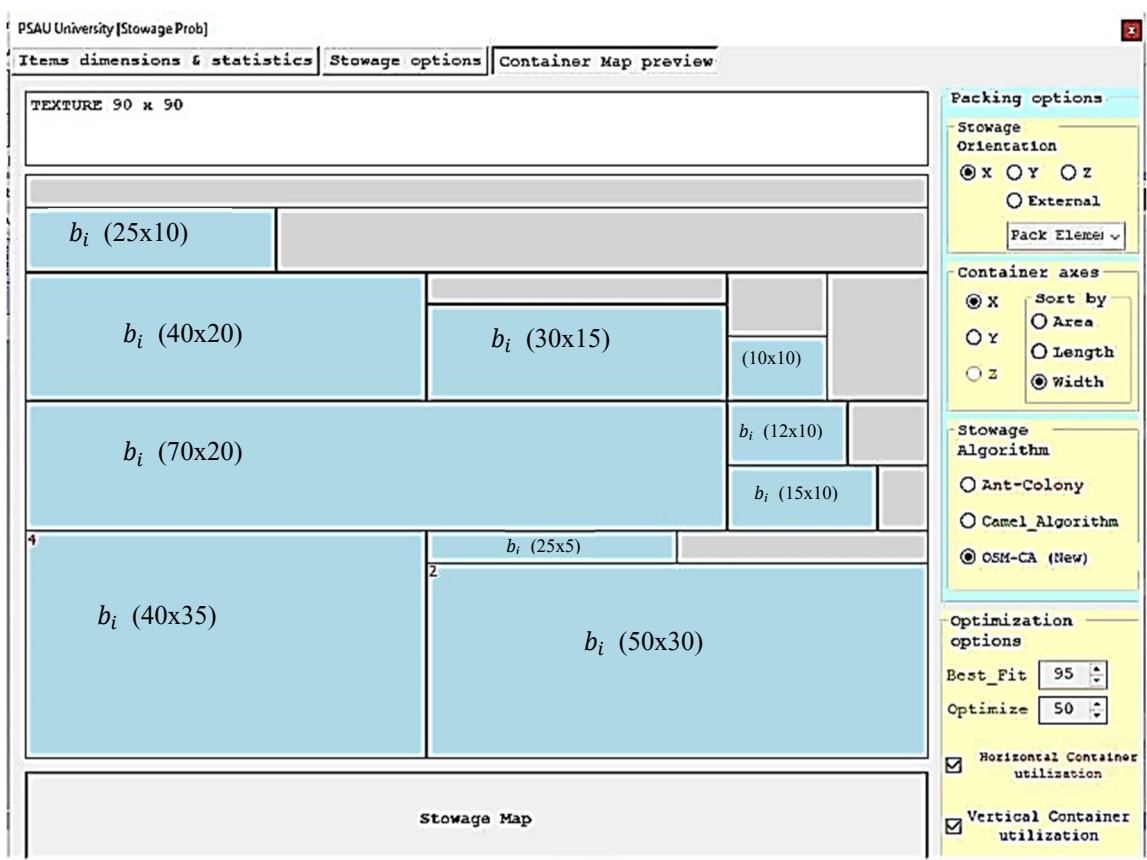

**Figure A7.** The stowage map by proposed algorithm optimization OSM-CA for first station on the distribution path.

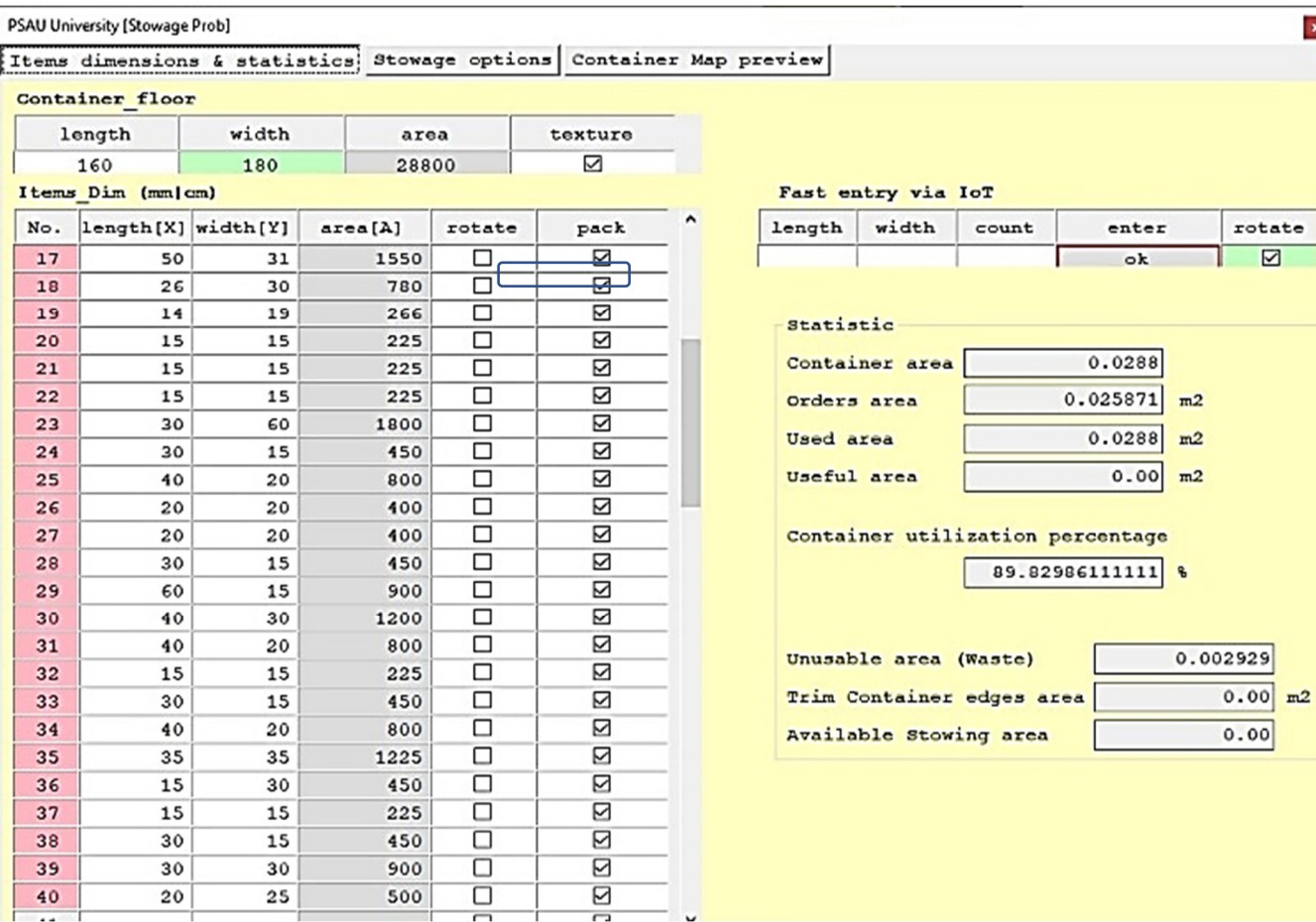

**Figure A8.** The container utilization by OSM-CA for third station on the distribution path.

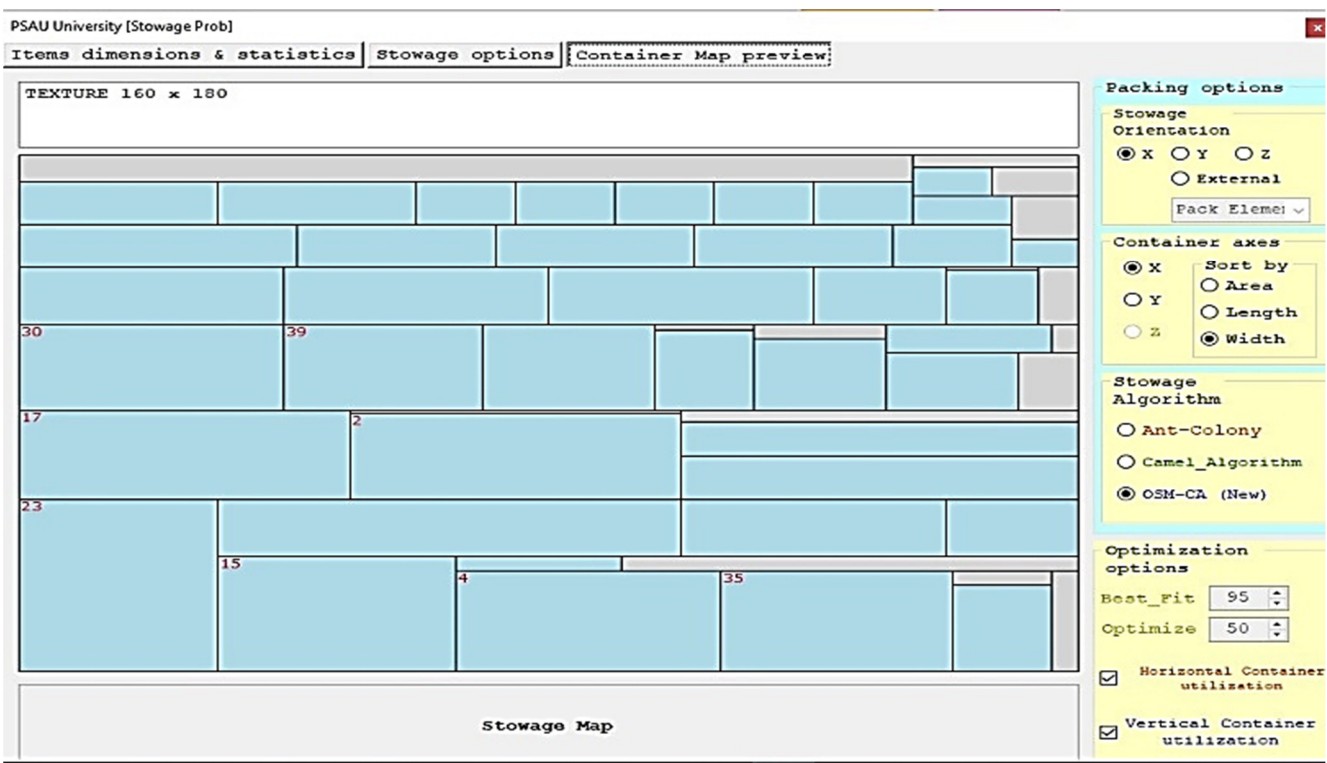

**Figure A9.** The stowage map by proposed algorithm optimization OSM-CA.

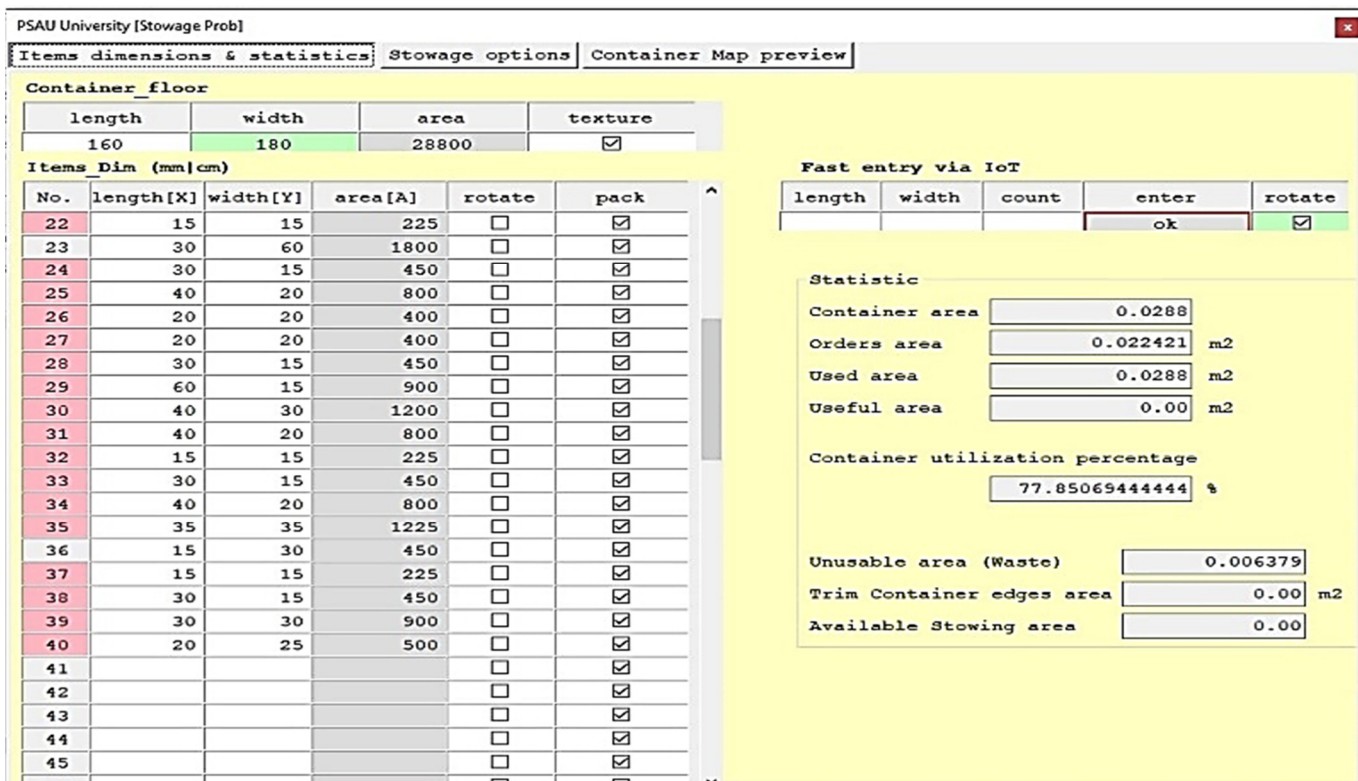

**Figure A10.** The container utilization by ant-colony algorithm for third station on the distribution path.

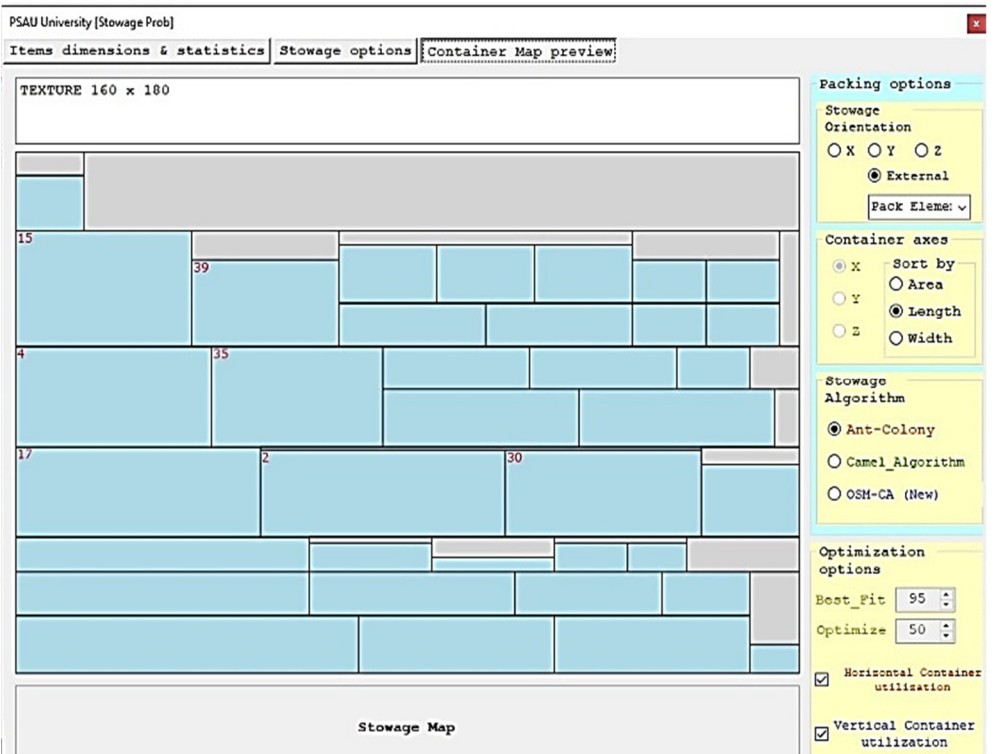

**Figure A11.** The stowage map by ant-colony algorithm for third station on the distribution path.

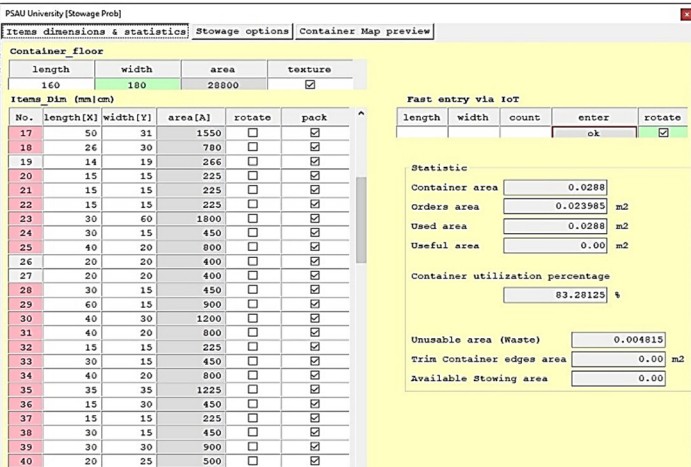

**Figure A12.** The container utilization by camel algorithm for third station on the distribution path.

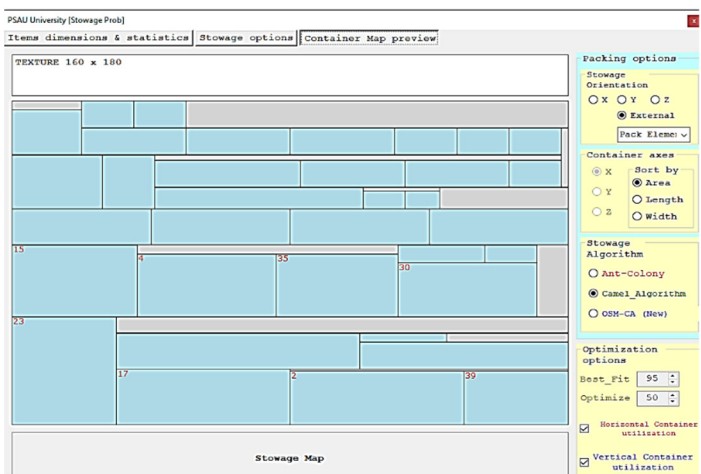

**Figure A13.** The stowage map by camel algorithm for third station on the distribution path.

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
