# Peer review of "The Lean-Branch-and-Bound Structure Effectiveness in Enhancing the Logistic Stowage Methodology for the Regular Shapes"

_processes, doi:10.3390/pr10112252_

Round 1

Reviewer 1 Report

The manuscript presents an IoT-based logistic cycle system using an algorithm called, Oriented Stowage's Map by Camel Algorithm (OSM-CA). The manuscript is well-written, and the system design and background is comprehensively presented. However, there are some concerns: 

1) Introduction does not highlight the contributions.

2) Related literature should be reviewed in detail by providing the differences from the present manuscript.

3) Figure 3 is not clear and explained in detail.

4) On page 4, the variables, x_l and y_l are not defined.

5) In “Assumption 5”, the referred diagram is not clear.

6) The numerical results need more detailed discussions to provide information.

7) The conclusions are too detailed, it should provide a summary of the manuscript, concluding remarks, and future work. 

Author Response

Dear reviewer

Thank you for reading and processing our manuscript, and we have benefited from your comments in enhancing our manuscript. The table below has the responses to your comments. 

Reviewer 2 Report

Paper 1922756

Areas with highlighted numbers. They must be rewritten, they are very long sentences, which make the understanding difficult.

Acronym LXX : L: Line; XX; number line

L17: Refers to C#?

Abstract: Not clear, generally an abstract does not contain examples, because it implies that what was previously stated is not clear.

L38: Review packing paper, there are contributions to the Strip Packing Problem, which can be referenced and applied to the case study.

L637: should review syntax

L38-640: Review bibliographic standard.

L641:Incomplete reference

L62:Be more quantitative, what do the authors understand by "Many Objectives".

L69: The methodology, having overlapping, becomes complex to follow and therefore to replicate, and traceability is lost. It is suggested to use more annexes and mind map, it is not clear how it is exposed.

L76: The figure is not clear. Analyze another option using pseudocode support and/or mind maps by phase.

L77-82: Describes only some explanations of the diagram in Figure 2. Citing the literature is favored, however, there is a lack of background information to replicate and understand what is presented.

L94-96: Summarize, the content in parentheses (), does not contribute since they are aspects of notation and training of the authors.

L99: More and better description of the heuristics is required, according to what has been reviewed in the literature it is similar to the classic placement algorithms.

L131: To have a repository with the instances, for replication of the experiment. The completeness of the experiment is lost.

L134: 8 hours of computation is not feasible for an operational case. Now why not 12 or 1 hour, there is a missing experiment for parameter calibration, which should be designed and clearly stated in the text.

L184: Lack of specification of the assemblies

L190: Watch out for the dot (.)

L197: Watch out for the dot (.)

L197-200: Revise wording

L333: Describe the dynamic procedure. Explain which aspects are dynamic and which are not (boundary conditions). To which probability distribution does it respond, you must make these data and results available.

L343: Why work with matrices, if the poor computational performance is known, is it the cause of the high computational times?

L477: They are extensive, non-standard procedures, they cannot be replicable, how do you guarantee that you explore the whole solution space, what are the exclusion criteria, how do you handle geometric constraints? How do you handle geometric constraints, what data structure do you use to support design decisions? Use standard aspects such as pseudocode.

Figure 13 and 14: no explanatory gloss, should be redone and explained appropriately.

Table 17: Show and make available the configuration of each solution.

Table 18: Use same measurement unit

L575: the section is analysis of results, not conclusions. It should be redone.

L816: Incorporate in the Leaks the part number, and whether or not it is rotated or rotated, as appropriate.

Author Response

Dear Reviewer

Thank you for reading and processing our manuscript, and we have benefited from your comments in enhancing our manuscript. The table below has the responses to your comments. 

Round 2

Reviewer 1 Report

The manuscript has been improved and revised to address the concerns raised by the Reviewer. It may be considered for publication in its current form. 

Author Response

Dear Reviewer 

Thank you for directing us to improve the manuscript

Yours Sincerely 

Reviewer 2 Report

Dear, please generate a repository where the instances are described with complete clarity.

In addition, it is requested to generate the layout of solutions.

It is requested to improve the quality of the figures, it is still poor.

We request the 10.X.X sections to be converted to pseudocode as well.

Urgent, the repository with the instances and it is required to make the executables available for performance testing.

Comparisons are missing to make the delivery of results more robust.

Author Response

Dear Reviewer

Thank you for directing us to improve the manuscript

The programmed code was uploaded with some examples of problems on google drive, please find the executable file and run it, and can check from the examples discussed in the Appendix-A

There are more tested cases for real situations for distributors such as 2B CO. and Ideal Standard for bathtubs and implemented via consultant project.

The link in the response file 

Yours Sincerely  

Round 3

Reviewer 2 Report

Dear colleagues, I refer to my comments:

Variables should be in italics, that includes figures.

The quality of the figures MUST be improved.

Authors should have a repository where both input and output files are available.

Authors should generate analytical conclusions.

There are some English sentences, very long, but grammatically correct.

Author Response

Dear Reviewer

We apologize for the lateness because we wait for one of the sponsors' firms "2B" to supply us with ten packages for different 10 customers over one working week. The number of customers serviced over targeted is ten customers, which means decreasing the number of trips by one weekly.